# Phytoplankton fatty acid proportions in the Canadian Arctic are strongly affected by temperature, salinity, and phosphate in late summer

Carlissa D. Salant[1][*], Jean-Éric Tremblay[2], Christopher C. Parrish[1][‡]

**1** Department of Ocean Sciences, Memorial University of Newfoundland, St. John's, Newfoundland and Labrador, Canada, **2** Département de biologie, Québec-Océan and Takuvik, Université Laval, Québec City, Québec, Canada

☉ These authors contributed equally to this work.
‡ CCP authors contributed equally to this work.
* cdsalant@mun.ca

## Abstract

Phytoplankton form the base of the food web and act as the main source of both energy, and ω3 essential fatty acids in marine ecosystems. In the Arctic, the effects of climate change are exacerbated compared to lower latitudes. Expected shifts in phytoplankton growth due to climate change in the Arctic are dependent on water physicochemistry, especially temperature, salinity, and nutrient concentrations. The goal of this investigation was to, 1) assess which Canadian Arctic (CA) regions were significantly different based on biomarkers of phytoplankton condition and community assemblage, and 2) identify which hydrographic properties and dissolved inorganic nutrient concentrations correlated with those biomarkers. Phytoplankton samples were collected from the CA in 2019 (July-September) and 2021 (August-October). CA regions were clustered based on similar hydrographic properties, and then lipid and FA profiles were compared. In 2019, there was variation in monounsaturated fatty acid (MUFA) proportions: cluster groups in very deep water, with higher temperature and lower oxygen saturation (Davis Strait stations) were significantly lower compared to cluster groups defined by lower salinity and light transmission (East Barrow Strait and Lancaster Sound stations) in both surface ($p = 0.001$) and sub-surface chlorophyll maximum waters (SCM)($p = 0.008$). In 2021, a year with more expansive coverage of the CA compared to 2019, Σω3% and polyunsaturated fatty acids (PUFA) were significantly higher in cluster groups containing Baffin Bay stations compared to the clusters defined by lower salinity (Beaufort Sea and the Canadian Arctic Archipelago stations) in both surface ($p < 0.001$) and SCM waters ($p < 0.001$). Across both years and both SCM and surface waters, DHA/EPA (two ω3 essential fatty acids) negatively correlated with phosphate. Overall, FA proportions in the CA are strongly affected by physiochemical conditions in late summer with MUFA responding mainly to physical changes, while PUFA responded heavily to phosphate changes.

**Data availability statement:** All relevant lipid, fatty acid, and nutrient data for each station are within the manuscript and its supporting Information files (S1 Dataset). All oceanographic files are available from the Amundsen Science database which has been attached as a URL (https://amundsenscience.com/data/data-access/).

**Funding:** This investigation was supported by Memorial University of Newfoundland, a grant from the Natural Sciences and Engineering Research Council of Canada (NSERC) to CCP and J-ÉT via the strategic network CHONe (Canadian Healthy Oceans Network), and a grant from the network center of excellence ArcticNet to J-ÉT and CCP (RGPIN-2017-04639 and RGPIN-2023-05858). The funders had no role in study design, data collection and analysis, decision to publish, or preparation of the manuscript.

**Competing interests:** The authors have declared that non competing interests exist.

## Introduction

The Arctic is warming faster compared to other oceans, with a concomitant decrease in Arctic sea ice thickness and an earlier ice melt in the spring [1–3]. A byproduct of this sea ice loss in the Arctic has been an increase in pelagic primary production by marine phytoplankton in shelf areas [4–6]. The warming and loss of sea ice thereby affects the timing and magnitude of pelagic Arctic phytoplankton blooms in the spring [7], resulting in a mismatch between available food sources and the occurrence of consumers [8]. Phytoplankton lipids and fatty acids (FAs) are an important component of overall marine food web health [9,10]. Lipids are the base from which energy, i.e., carbon and essential nutrients, is transferred to primary consumers and, subsequently, to secondary consumers and predators [11,12]. Various studies have examined how this transfer of energy pathway may be altered due to climate change. While some studies suggested no initial deleterious effect on primary consumer health with decreased essential nutrient value of primary producers [13,14], other studies have suggested that over the long term, a decrease in primary producer essential nutrient value would negatively affect consumer health at multiple food web levels [15,16].

Lipids provide a more efficient energy source in the marine ecosystem compared to carbohydrates and proteins and can be divided into two main acyl (R–C=O) groups: polar and non-polar lipids. Generally, polar lipids, such as phosphoglycerides (PL) and acetone-mobile polar lipids (AMPL) are important structural components of cell membranes and non-polar lipids (triaclyglycerols, TAG; wax esters, WE) act as energy stores. The chemical structure of acyl lipids includes FAs, which along with glycerol, act as the building blocks of many lipids. Composed of carbon chains, often with ethylenic bonds (–C=C–), FAs are commonly designated as saturated (SFA), monounsaturated (MUFA), and polyunsaturated (PUFA) based on the number of double bonds. These molecular variations affect the physiology of organisms, are important to their functions within marine food web, and are useful as diet biomarkers [17–19]. Additionally, some FAs are synthesized almost exclusively by primary producers, providing the only means by which consumers can gain these essential nutrients [10]. These are deemed essential fatty acids (EFAs) and include linoleic acid (LA; 18:2ω6), alpha-linolenic acid (ALA; 18:3ω3), arachidonic acid (ARA; 20:4n6), eicosapentaenoic acid (EPA; 20:5ω3), and docosahexaenoic acid (DHA; 22:6ω3).

In the Canadian Arctic (CA), the spring phytoplankton bloom onset and magnitude tend to be tightly controlled by physical and chemical variables including sunlight, seawater and sea-ice properties, nutrient availability, and input of freshwater [20–22]. While especially vital for phytoplankton growth and ecosystem health in the spring, these physical and chemical variables are also important in the Arctic as spring transitions to summer, given that low irradiance and low nutrient concentrations tend to characterize the upper water column in late summer and throughout the winter [23]. Overall, these physical and chemical variables control phytoplankton chemical composition, altering lipid and FA composition as the seasons change. Studies with cultured algae have found that when temperature is the sole variable, TAG, SFA, and MUFA increase proportionally in phytoplankton [24–26]. Important EFAs also

correlate with temperature; ALA, EPA, and DHA decrease with increasing temperatures [27–29], whereas TAG proportionally increases [30]. Primary producer growth and EFA synthesis require light to drive the onset of the spring bloom in the Arctic [31,32]. Depending on growth stage of individual phytoplankton species, light levels can influence lipid production [8,33–35]. Decreases in salinity can alter phytoplankton species composition [36,37,38], indirectly leading to changes in FA profiles [17,39,38].

Dissolved inorganic nutrients also play a role in primary production and lipid synthesis, especially after the initial spring bloom. A lack of nitrogen can limit primary production [40] and further decrease phytoplankton nutritional quality [41,42]. Nitrate availability correlates with the magnitude of the later, subsurface, summer phytoplankton blooms in the Arctic [33]. Phosphate limitation within an individual cell increases both SFA and MUFA, while decreasing PUFA and increasing TAG at the expense of PL [43]. Silica limitation, predominantly affecting diatoms, can affect both their division rates and lipid content, and this limitation leads to an increase in TAG and higher proportions of SFA and MUFA [44,45]. In conjunction with physicochemical properties, such as temperature, a decrease in overall nutrient concentrations can increase lipid production [46]. In addition to physical and chemical variables, phylogeny also shapes phytoplankton community composition [39], and in the Canadian Arctic phylogeny varies by region [47]. Although it is difficult to discern using FA alone if a shift in lipids is due to individual species changes or shifts in phytoplankton composition, a combination of phytoplankton phylogeny, seawater properties, seasonal taxonomic succession, region, and nutrient availability likely defines the regionally specific FA profiles in the Canadian Arctic (CA) [29,31,22,48].

The CA extends from the eastern edge of the Beaufort Sea, through the Canadian Arctic Archipelago, into Baffin Bay, and down into the northern part of the Labrador Sea. The Canadian Arctic Archipelago is affected by inflows from the Beaufort Sea to the west, Baffin Bay to the east, and the Arctic Ocean from the north. In Baffin Bay two different water masses affect the area; the relatively cold and fresh waters from the high Arctic descending south, and the warmer Atlantic waters moving norward through the West Greenland Current [49,50]. These ocean circulation patterns give rise to varying net primary production (NPP) throughout the CA. In the Beaufort Sea, production has increased while in the northern Baffin Bay and North Water Polynya, NPP has decreased as a result of surface freshening and stratification [51]. Given the complex water circulation patterns and changing NPP values, it is expected that any change in hydrographic properties will cause large changes in phytoplankton growth, abundance, taxonomy, and FA profile [52]. Thus, advancing our knowledge of the relationship between environmental drivers and phytoplankton in the CA will ultimately help elucidate in which regions productivity may increase or decrease.

To focus on environmental drivers, we compiled regional seawater profiles and investigated which hydrographic properties influenced FA condition biomarkers, as well as phytoplankton assemblage biomarkers. Whereas individual metrics can be informative, biomarkers can provide a more holistic view of an entire lipid and FA profile. Total lipids, lipid class ratios, and FAs focusing on Σω3 and ΣPUFA patterns were used as biomarkers to assess phytoplankton condition and nutritional quality, and FA summations and ratios were used to infer plankton population composition [18,10,53]. Beyond DHA and EPA, the broader Σω3 profile of phytoplankton can also be evaluated when determining a phytoplankton population nutritional value [10]. FA biomarkers used to infer assemblage can discriminate among typical microplankton groups, including bacteria, diatoms, and flagellates [10,47,54]. Seston assemblages can also be categorized based on source, i.e., the location in the marine environment from which the organism or its detritus originated [52,53], and FAs can distinguish among organic matter sources that are more coastal (riverine/fluvial/terrestrial) like seagrasses, kelp, vascular land-plants, and green macroalgae [55–57]. This capacity is especially important because throughout the water column much of the suspended matter that is photosynthesizing can originate from multiple sources in the marine environment.

Here, we investigate the lipid and FA profiles of surface and sub-surface chlorophyll maximum (SCM) particulate matter samples collected from the CA, ranging from the Beaufort Sea to Baffin Bay, in both 2019 and 2021. We first correlated the hydrographic properties with lipid classes, lipid condition metrics including total lipids, FAs, and FA condition and

phytoplankton assemblage biomarkers. Secondly, cluster analyses were performed within each year and water column depth to delineate stations into Arctic clusters based on similar hydrographic properties. We finally compared the lipid and FA profiles among those resulting clusters to answer our main question of: which environmental conditions are most important in addressing FA in the CA?

## Materials and methods

### Phytoplankton sampling

Seawater for both filtered phytoplankton and nutrient analyses were collected in the CA between July 8th – September 3rd 2019, and between August 15th – October 4th 2021, from the CCGS *Amundsen*. In 2019, samples were retrieved from 33 stations and in 2021, samples were retrieved from 42 stations (Fig 1; S1 Table). Niskin bottles collected water for both filtered particulate matter (phytoplankton) and water analysis. During both years, surface and SCM phytoplankton samples were taken by filtering 3–5 L of seawater through 47-mm GF/C filters at each station. The SCM was determined based on the highest fluorescence values within a vertical profile using affixed biogeochemical sensors. Filters were pre-combusted at 450°C for 12 h and stored in aluminum foil. Once collected, samples were kept frozen at −20°C on the ship before being transferred to the home laboratory freezer (−20°C) until analysis.

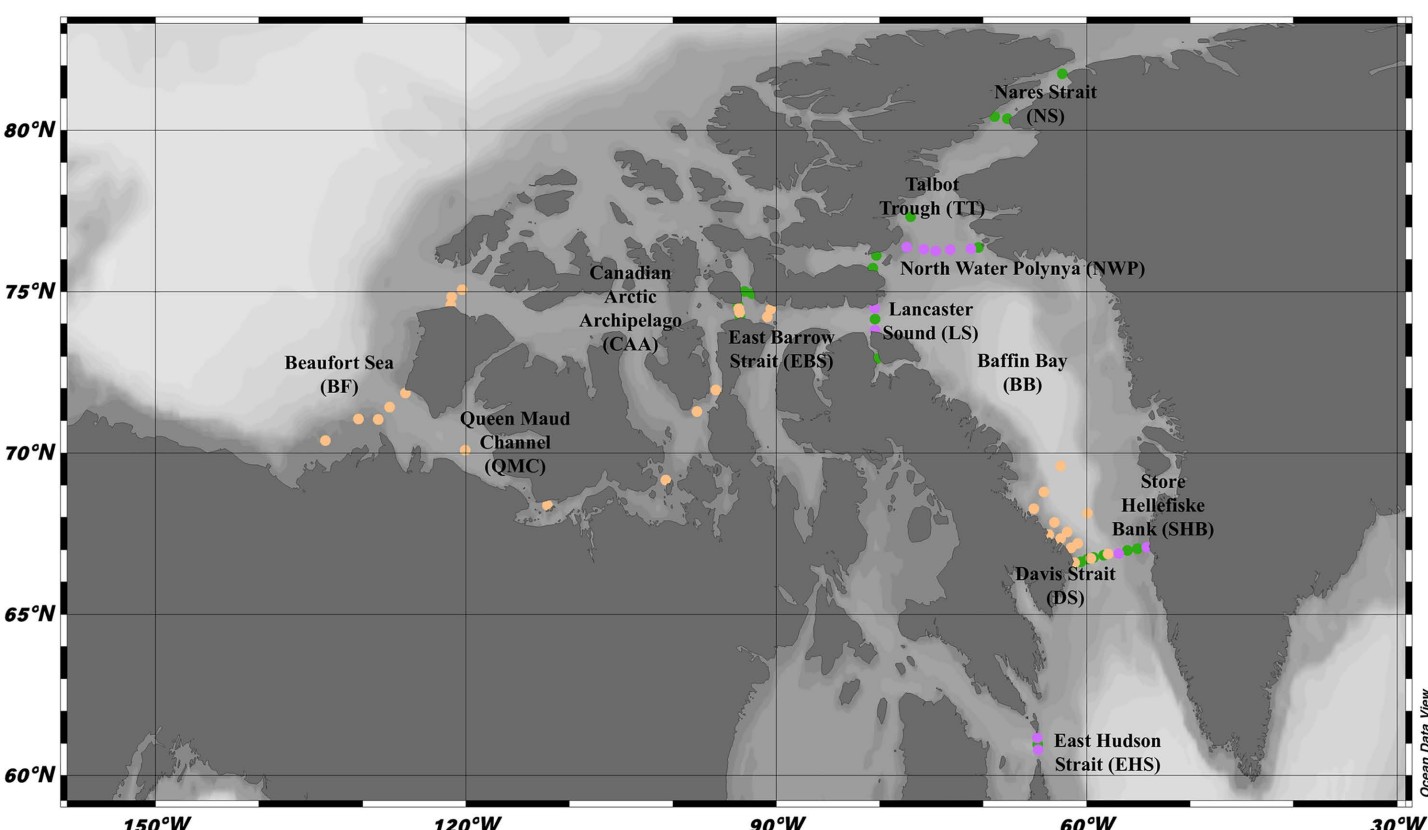

**Fig 1. Canadian Arctic survey map.** Survey map representing stations for 2019 (green, n = 33), 2021 (orange, n = 42), and stations sampled in both years (purple, n = 11 stations from 2019 were sampled again in 2021). Regions are in bold. See Table 1 for detailed locations. Map created with permission (Schlitzer, Reiner, Ocean Data View, https://odv.awi.de, 2021.).

**Table 1. 2019 OceanMet summary.**

| | Regions within each cluster | Stations in cluster (*n*) | Shorthand area name | Description | Same region? | Significantly higher | Significantly lower | Shorthand ocean metric description |
|---|---|---|---|---|---|---|---|---|
| **Surf.** | East Hudson Strait | 3 | EHS | All EHS stations | Yes | Nitrate | | high nitrate |
| | Store Hellefiske Bank/North Water Polynya | 4 | SHB/NWP-East | All SHB stations, plus the NWP station closest to the Greenland coast | No | Temperature | Oxygen saturation | high temp/low oxygen sat. |
| | Davis Strait/North Water Polynya | 3 | DS/NWP | The DS and NWP stations in the middle of their relative channel | No | Temperature, bottom depth | Oxygen saturation | high temp & oceanic/low oxygen sat. |
| | Davis Strait west/Nares Strait | 2 | DS-West/NS | Coastal DS and NS station | No | Oxygen saturation | | high oxygen sat. |
| | Davis Strait west/Nares Strait-North | 2 | DS-West/NS-North | Coastal DS station and the northern NS station | No | Oxygen saturation | | high oxygen sat. north |
| | North Water Polynya/North Water Polynya-West | 4 | NWP/NWP-West | NWP stations closest to the Canadian coast | Yes | Average dissolved oxygen, nitrate sensor, phosphate | | average nutrients |
| | East Barrow Strait/North Water Polynya-West/Talbot Trough | 7 | EBS/NWP-West/TT | All EBS stations, with the addition of northern NWP inlet stations | Yes | Oxygen saturation | Salinity | high oxygen sat./low salinity |
| | Lancaster Sound/Davis Strait | 4 | LS/DS | Lancaster Sound stations and a coastal DS station | No | Bottom depth | | high bottom depth (oceanic) |
| | Lancaster Sound/Nares Strait | 2 | LS/NS | Northern LS and NS station | No | | Salinity, light transmission | low salinity & light |
| **SCM** | East Hudson Strait | 3 | EHS | All EHS stations | Yes | Average bottom depth, fluorescence, | | average |
| | Store Hellefiske Bank | 3 | SHB | All SHB stations | Yes | Temperature | Oxygen saturation, nitrate sensor, phosphate | high temp/low oxygen & nitrate sensor & phosphate |
| | Davis Strait/Lancaster Sound-North | 5 | DS/LS-North | The more eastern DS stations and the northern LC station | No | Light transmission | | high light |
| | Lancaster Sound, Nares Strait, North Water Polynya East and West | 8 | LS/NS/NWP | Middle LS, all NWP east and west stations, and Nares Strait stations | No | Average bottom depth | | epipelagic |
| | Davis Strait, Lancaster Sound south, East Barrow Strait, North Water Polynya-West Inlet, Talbot Trough (TT) | 11 | EBS/NWP/TT | Coastal DS station, southern LS station, NS north station, all EBS stations, NWP west inlet station, and TT station | No | Dissolved oxygen | Salinity | high dissolved oxygen/low salinity |

A descriptive summary of the OceanMet groups produced by hierarchical cluster analysis (SIMPROF) based on seven ocean metrics in surface (S1-A Fig) and sub-surface chlorophyll maximum (SCM)(S1-B Fig) waters that were collected between July 8th – September 3, 2019. Shorthand names include East Hudson Strait (EHS), Store Hellefiske Bank (SHB), North Water Polynya (NWP), Davis Strait (DS), Nares Strait (NS), Lancaster Sound (LS), East Barrow Strait (EBS), and Talbot Trough (TT).

To ensure enough material was gathered for both lipid class and FA analyses, either one, two, or three filters were used. In 2019, one filter was used to represent each station for both surface and SCM. In 2021, three filters combined were used to represent each station for surface and SCM samples at stations that coincided with identical 2019 stations. For the 9 stations in the Beaufort Sea in 2021, two filters combined were used to represent both surface and SCM samples. Overall, each station had a replicate number of *n* = 1.

## Hydrographic and chemical properties

To assess hydrographic and chemical properties of the seawater, sensors were mounted on the same rosette as used for seston sampling to provide vertical profiles of depth, temperature (Sea-Bird Scientific SBE-911 CTD), dissolved oxygen (Sea-Bird Scientific SBE-43, calibrated onboard against Winkler titrations) [58], chlorophyll fluorescence (Seapoint), nitrate sensor, and light transmission. Samples for nutrients were collected from the Niskin bottles with syringes, filtered through a Whatman GF/F filter mounted in a Swinnex filter holder, and rapidly stored in the cold in acid-cleaned polyethylene tubes. The nitrate sensor did not take into account dissolved nitrate and therefore remains unitless, but nitrate was explicitly calculated later (see below). While fluorescence is used as a measure of chlorophyll $a$ (mg/m$^3$), this value was not adjusted in the lab and therefore remain as volts (V).

Nutrient concentrations for nitrate + nitrite, nitrite, phosphate, and silicate were measured colorimetrically using a Bran and Luebbe AutoAnalyzer III within a few hours of collection. Analytical detection limits were 0.03 μM for nitrate (obtained by the difference between nitrite and nitrate + nitrite), 0.02 μM for nitrite, 0.05 μM for phosphate, and 0.1 μM for silicate. For simplicity, nitrate in text hereafter refers to the combined concentrations of nitrate+nitrite, noting that the latter represents a small fraction of the former (not to be confused with nitrate sensor which is the raw output variable). Ammonium concentrations were determined manually using the method of [59]) with a detection limit of 0.02 μM.

## Total lipids, lipid classes, and fatty acids

Each filter was weighed before storage upon arrival at the home laboratory. The mass of the sample was calculated by subtracting the mass of clean, pre-combusted filters (n = 3). An assumption of 20% water mass was used to account for the frozen seawater on the sample for total lipid concentrations (JAD Ángel-Rodríguez, August 2020, personal communication). At the onset of chemical analysis, filters were put in glass, pre-combusted vials; each sample contained either one, two, or three filters (S1 Table). Vials were filled with 2 mL of chloroform, flushed with nitrogen, capped and sealed with Teflon tape, and stored at −20°C until extraction.

Lipids were extracted using a modified chloroform:methanol:chloroform extracted-water (1:2:1) mixture [60,61]. A homogenizer ground the samples into a uniform pulp. The lipids were extracted using a double pipetting method and were sonicated for 4 mins and centrifuged for 3 mins at 3000 rpm. This procedure was repeated 3 more times. The extracts were flushed with nitrogen, capped and sealed with Teflon tape, and stored at −20°C until lipid class analysis. Thin-layer chromatography with flame ionization detection (TLC-FID, Mark V Iatroscan, Iatron Laboratories) determined total lipids and lipid class proportions [61]. Lipid classes were separated using a four-step development procedure on silica gel Chromarods. The first development contains hexane:ethyl ether:formic acid (98.05:1:05) and separates hydrocarbons (HC), wax and steryl esters, ethyl/methyl esters (EE/ME), and ethyl/methyl ketones (EK/MK). The second development contains hexane:ethyl ether:formic acid (79:20:1) and separates triacylglycerols (TAG), free fatty acids (FFA), and sterols (ST). The third and fourth stages contain acetone (100%) and chloroform:methanol:chloroform-extracted-water (5:4:1) and separate the more polar classes, the acetone-mobile polar lipid (AMPL) and phospholipid (PL) classes. Nine Sigma Chemical Inc. standards were used for FID calibration and analysis was conducted using Peak Simple software (version 4.89).

A portion of the lipid extracts was placed in pre-combusted vials. A 4.5 mL mixture of $H_2SO_4$-MeOH was added to the vial, which was then flushed with nitrogen, capped and sealed with Teflon tape, and then heated at 100°C for 1 hour. After cooling for 5 mins, 1.5 mL of hexane was added, samples were vortexed for 5 mins, prior to transferring the upper layer containing the FAs was transferred into a clean vial. Samples were again flushed with nitrogen, vortexed, capped and sealed with Teflon tape prior to storage at −20°C until analysis. Fatty acid methyl ester (FAME) were analyzed using a gas chromatograph (GC-FID) [53,62] with an autosampler and DB WaxPlus GC column (30 x 0.25 mm x 0.15 μm). Fatty acid peaks were identified and integrated using relative retention times compared to FAME standards (Nu-Chek Prep).

## Statistical analyses

Twelve hydrographic and chemical variables (temperature, salinity, bottom depth, fluorescence, light transmission, oxygen saturation, dissolved oxygen, nitrate sensor, nitrate, silicate, phosphate, and ammonium) were evaluated for skewness (e.g., Shapiro–Wilk test), homogeneity of variances (e.g. Levene's test), and collinearity (variance inflation factors, VIF) prior to analysis. Variables with skewed distributions were log-transformed. Concentration variables were also log-transformed to stabilize variances before statistical analyses. Total lipids are presented as mg/g wet weight (WW), while the rest of the lipid classes and FAs were are expressed as proportions (% total). Analyses were restricted to lipid classes and FA contributing >1% of the total across both years, yielding eight lipid classes (of 14 possible) and 17 FAs (of 74 identified). Lipid metrics of phytoplankton condition and nutritional quality included total lipids, TAG/PL, TAG/ST. FA biomarkers of phytoplankton condition and nutritional quality included PUFA/SFA, $\Sigma\omega3$, and $\Sigma\omega6$. Phytoplankton assemblage biomarkers were classified as bacterial ($i$15:0, $ai$15:0, 15:0, 15:1, $i$16:0, $ai$16:0, $i$17:0, $ai$17:0, 17:0, 17:1, and 18:1$\omega$6), diatom (16:1$\omega$7/16:0), flagellate ($C_{18}$PUFA/$C_{16}$PUFA), DHA/EPA (dinoflagellates) or coastal margin (18:3$\omega$3 + 18:2$\omega$6). An additional metric (DHA + EPA) is included in the analyses, despite the overlap with DHA/EPA and $\Sigma\omega3$.

Analyses were conducted separately for four groups defined by year and depth (2019 surface, 2019 SCM, 2021 surface, 2021 SCM). Correlation analyses were performed among all hydrographic variables and lipid classes, FAs, and biomarkers, and the number of significant correlations across groups was summarized. Multivariate analyses were conducted using PRIMER v7 (Primer-E Ltd) [63,64]. Hierarchical cluster similarity profile permutation tests (SIMPROF; Type 1, 999 permutations) were conducted separately for each year and depth using Euclidean distance. Outliers identified by SIMPROF were excluded. Final clusters, based on oceanographic properties only (excluding nutrients) were termed "OceanMet" groups. Nutrient profiles were examined subsequently in relation to these groups; if nutrients were included in the cluster permutation tests, the groups yielded too many outliers. The advantage of creating these groups was that we were able to specifically test for differences based on comparisons among different hydrographic properties and their FA profiles. Our aim was not to explicitly explore geographical differences, although sometimes our resulting cluster groups did contain stations from the same region.

One-way permutational multivariate ANOVA (PERMANOVA; Type III SS, 999 permutations) with Bray–Curtis similarity was used to test for differences among OceanMet groups, followed by pairwise tests. Principal coordinates analysis (PCoA) was used to visualize relationships among OceanMet groups, lipid metrics, and biomarkers (TAG/ST excluded due to missing values), using log(x + 1)-transformed data and Bray–Curtis similarity. Finally, univariate comparisons of lipid classes and FAs among OceanMet groups were tested using one-way ANOVA with Tukey's HSD or, when assumptions of normality/variance were violated, non-parametric Kruskal–Wallis tests (Minitab v21).

# Results

## 2019 Sample Collection

**Surface. 2019 Surface hydrographic properties.** Between July 10 – September 3 2019, 33 surface and 33 SCM phytoplankton samples were collected concurrently from stations in the CA (Fig 1; S1 Table). Cluster similarity profile analysis (SIMPROF) by seven oceanic and six inorganic nutrient measurements yielded nine OceanMet groups in 2019 surface waters: 1) high nitrate, 2) high temperature/low oxygen saturation, 3) high temperature and depth/low oxygen saturation, 4) high oxygen saturation, 5) high oxygen saturation north, 6) comparatively average, 7) high oxygen saturation/low salinity, 8) oceanic, and 9) low salinity and light (S1-A Fig; S1 Table and Table 1). Of the 33 stations, two were not included in any cluster and were removed from further analysis. Of the nine resulting OceanMet groups in 2019 surface waters, only one group emerged representing a singular region: East Hudson Strait. Seven of the nine clusters included Davis Strait and North Water Polynya areas, including clusters that varied in oxygen saturation (i.e., low and high oxygen saturation in the surface). Of the seven ocean metrics and five inorganic nutrient measurements averaged across the nine clusters, all but ammonium varied significantly among clusters (S2–S3 Fig).

## 2019 Surface lipids and fatty acids

Of the eight lipid classes, FFA was significantly lower, and polar lipids were significantly higher in the high temperature/ low oxygen saturation OceanMet group compared to the high oxygen saturation/low salinity OceanMet group, whereas none of the three lipid condition markers (total lipids, TAG/ST, TAG/PL) differed (S2 Table). The SFAs were the highest, proportionally, among the saturation groups, ranging between 27.1–62.9%. The MUFAs ranged between 15.9–41.8%, and PUFAs ranged between 18.4–49.2% (Fig 2). Of the 17 FA above 1%, eight FAs including steric acid (18:0), arachidic acid (20:0), palmitoleic acid (16:1ω7), hexadecatetraenoic acid (16:4ω1), 18:2ω6, 18:3ω3, steridonic acid (18:4ω3), and 22:6ω3, differed significantly among ocean data groups and of the three saturation summations, ΣMUFA, differed significantly among ocean data groups (Fig 2; S2 Table).

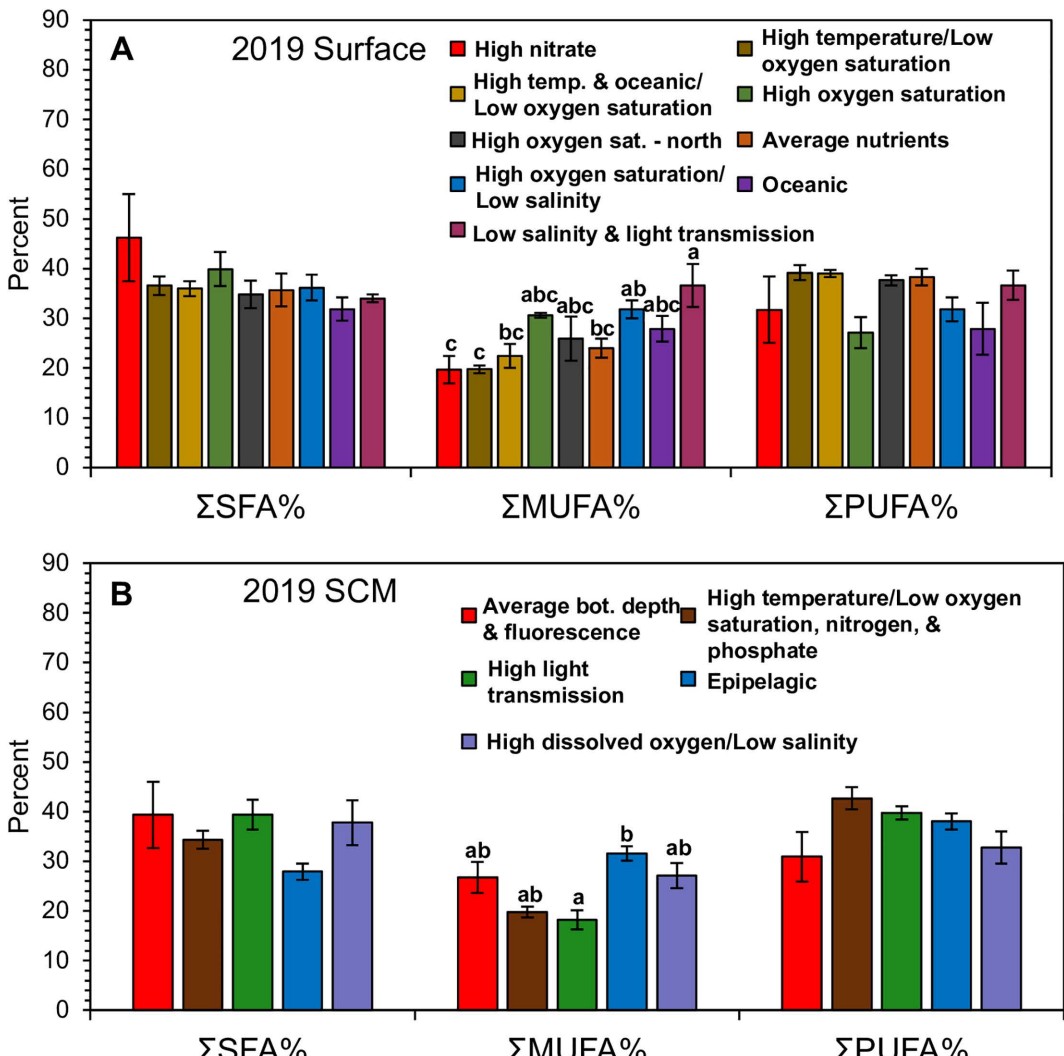

**Fig 2. 2019 Phytoplankton fatty acid summary.** Summary of the phytoplankton fatty acid sums and biomarkers, gathered from surface (A) and subsurface-chlorophyll maximum (SCM) (B) waters from July 10 – September 3, 2019. Fatty acid sums are grouped by OceanMet group and averaged (±SE) across saturated fatty acids (SFA%; no sig. differences), monounsaturated fatty acids (MUFA%), and polyunsaturated fatty acids (PUFA%; no sig. differences). Tukey comparisons (lowercase letters) indicate significant (p ≤ 0.05) differences among groups.

Of the eight FA biomarkers, five (Σω6, DHA/EPA, bacterial%, diatom, and coastal origin%) differed significantly among ocean data groups (S2 Table). Generally, these differences resulted from a higher proportion of bacteria and coastal margin biomarkers in the high temperature and low oxygen saturation OceanMet group compared to the high oxygen saturation and low salinity, oceanic OceanMet groups (S2 Table).When plotted in non-parametric space utilizing the eight FA biomarkers (PUFA/SFA, Σω3, Σω6, DHA/EPA, bacterial, diatom, flagellate, coastal origin) and the three lipid metrics (total lipids, TAG/PL, TAG/ST), the overall distance among OceanMet groups was significant (PERMANOVA; df = 8, pseudo-F = 2.16, p = 0.002). Pairwise PERMANOVA results showed that the significant difference among all OceanMet groups was driven mainly by pairwise differences among groups separated by temperature, salinity, and oxygen differences (S3 Table). In two axes, the PCoA explained 73.1% of the variability among the OceanMet groups using the eight FA biomarkers and two lipid (total lipids and TAG/PL; TAG/ST was not used due to insufficient ST values) metrics (Fig 3).

## 2019 Surface correlations

Of the seven oceanic measurements, all significantly correlated with at least one lipid class, lipid metric, FA, or FA biomarker, mainly positively (S4 Table). Salinity significantly correlated with the most lipid classes and lipid metrics (five of 11), mainly negatively, although it strongly correlated, positively with the polar lipid class. Of the lipid classes and lipid metrics, the lipid class, PL, had the most significant correlations with the seven oceanic measurements (four of seven), equally positive (temperature and salinity) and negative (dissolved oxygen and oxygen saturation). Temperature significantly correlated with the most individual FAs (five of 17), mainly positively, and FA biomarkers (five of eight), mainly positively (Fig 4). Temperature was most strongly correlated (correlation coefficient > 0.5), negatively, with the FAs 16:1ω7 and 16:4ω1 and the FA flagellate biomarker, while also strongly correlating, positively, with DHA/EPA. The FA 16:1ω7 was strongly correlated, negatively, with temperature and salinity and strongly correlated, positively, with dissolved oxygen and oxygen saturation (S4 Table).

Of the five inorganic nutrient measurements, all correlated with at least one lipid class, lipid metric, FA, or FA biomarker, mainly negatively (S4 Table). The nitrate sensor correlated with the most lipid classes and metrics (four of 11), mainly negatively. The nitrate sensor also correlated with the most FAs and FA biomarkers (16 of 28), mainly negatively. The nitrate sensor strongly correlated, negatively, with 20:0, 18:1ω7, 18:2ω6, 18:3ω3, 18:4ω3, DHA/EPA and the diatom biomarker strongly correlated, positively, with 16:1ω7, 16:4ω1, 22:1ω9, ΣMUFA, and the flagellate and coastal margin FA biomarkers (S4 Table). Eight FAs and FA biomarkers correlated with the most inorganic nutrient measurements (three of five), including 20:0 (mainly negatively), 16:1ω7 (mainly positively), 16:4ω1 (positively), ΣMUFA (mainly negatively), 18:3ω3 (mainly negatively), 18:4ω3 (mainly negatively), 20:5ω3 (mainly negatively), 22:6ω3 (negatively), and the diatom FA biomarker (mainly negatively) (Fig 4; S4 Table).

## SCM

**2019 SCM hydrographic properties.** Cluster similarity profile analysis (SIMPROF) of seven oceanic measurements created five OceanMet groups in 2019 SCM waters: 1) comparatively average, 2) high temperature and low light, nitrate sensor, and phosphate, 3) high light, 4) average depth, and 5) high dissolved oxygen and low salinity (S4 and S5 Figs; Table 1 and S1 Table). Of the 33 stations, three did not cluster with any group and were removed from further analysis; two of those three were the same stations removed from the surface waters. Of the five resulting OceanMet groups in 2019 SCM waters, two regions were also defined by hydrographic properties: East Hudson Strait (EHS) and Store Hellefiske Bank (SHB). Two of the five OceanMet clusters included the North Water Polynya areas. Of the seven ocean metrics and five inorganic nutrient measurements averaged across the five clusters, all varied significantly among clusters (S2–S3 Fig).

**2019 SCM lipids and fatty acids.** Of the eight lipid classes, TAG was significantly higher in the average OceanMet group compared to the high temperature/low oxygen, nitrate sensor, and phosphate OceanMet group, while one of the

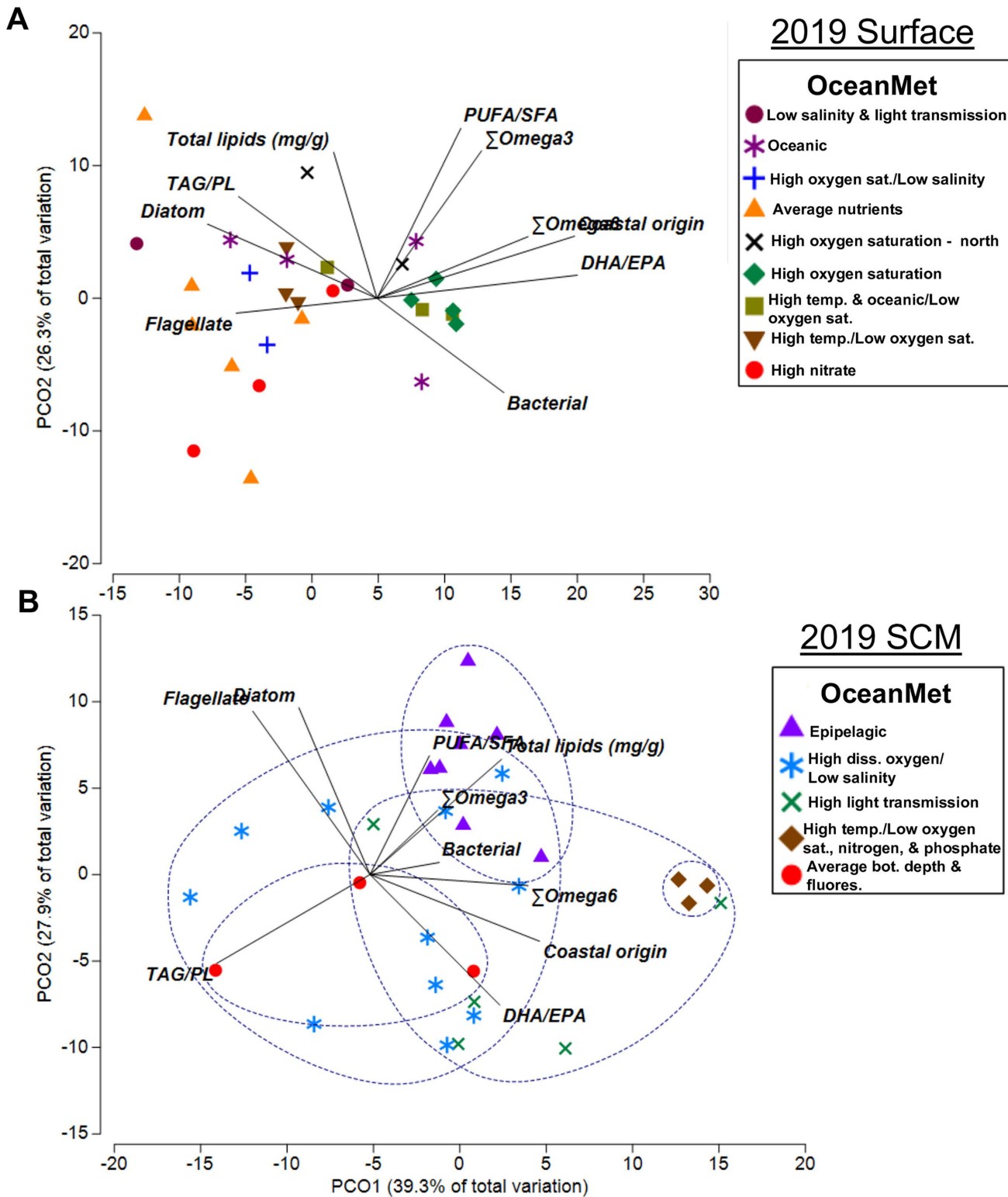

**Fig 3. 2019 PCoA biplot.** Fatty acid and lipid biomarkers plotted on a principal coordinate analysis (PCoA) plot and grouped by OceanMet (dotted blue lines) for surface (A) and subsurface-chlorophyll maximum (SCM) (B) waters from July 10 – September 3, 2019. OceanMet groups are also listed by shorthand area name (Table 1). Biomarkers include total lipids (mg/g), PUFA/SFA, ω3, ω6, bacterial ($i$15:0, $ai$15:0, 15:0, 15:1, $i$16:0, $ai$16:0, $i$17:0, $ai$17:0, 17:0, 17:1, and 18:1ω6), diatom (16:1ω7/16:0), flagellate ($C_{18}$PUFA/$C_{16}$PUFA), DHA/EPA (dinoflagellates) and coastal margin (18:3ω3 + 18:2ω6).

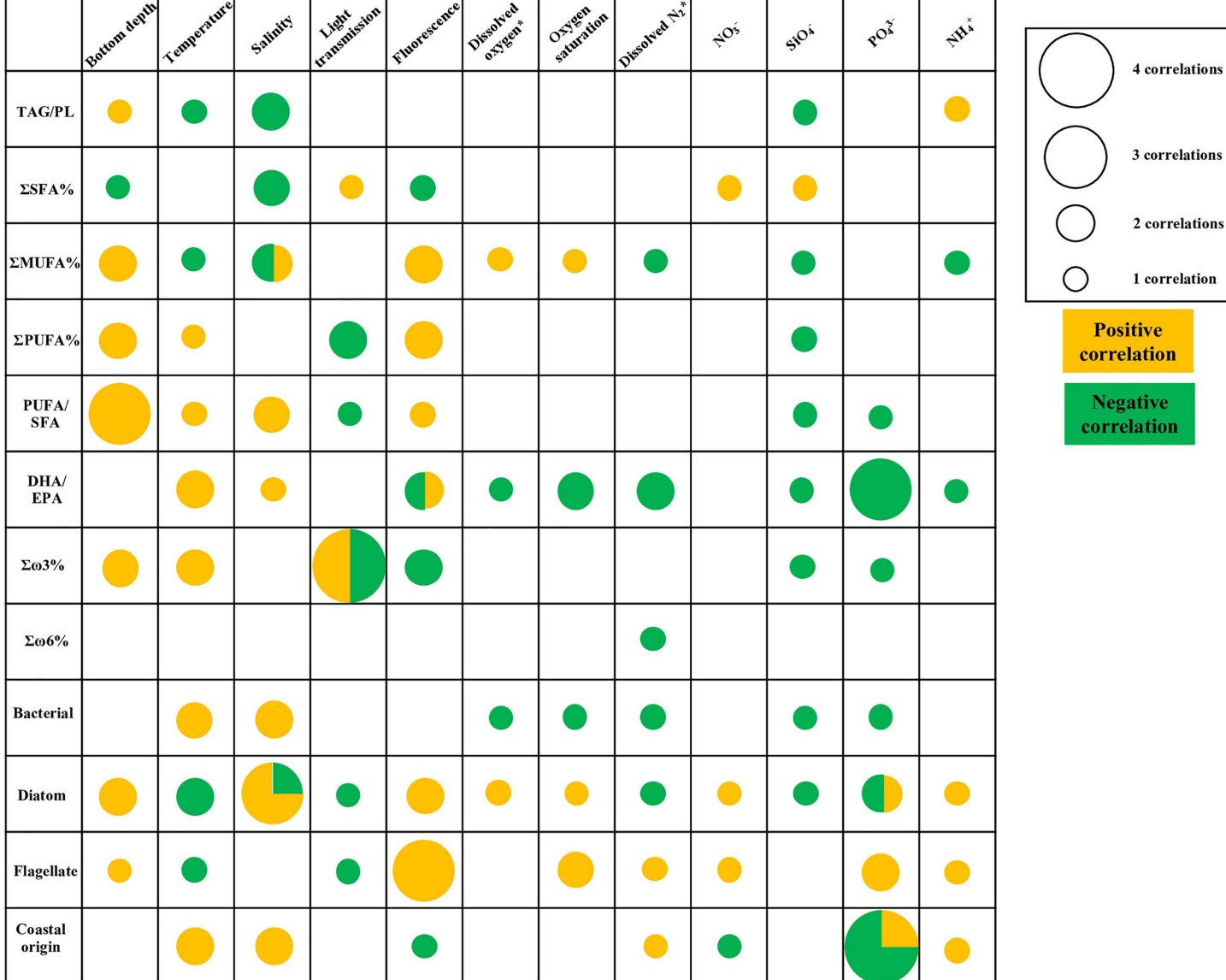

**Fig 4. Correlation bubble matrix.** Matrix depicting positive (orange) and negative (green) correlations among oceanic measurements (bottom depth, temperature, salinity, light transmission, dissolved oxygen, oxygen saturation, nitrate sensor, nitrate + nitrite, silicate, phosphate, and ammonium), and a lipid class condition metric, fatty acid saturation summaries, and fatty acid biomarkers. Bubble size represents the number of significant correlations, ranging from one to four, found across either 2019 or 2021, or across surface or subsurface chlorophyll maximum (SCM) waters. Supplemental S4 Tables and S9 separate correlations into 2019 Surface, 2019 SCM, 2021 Surface, and 2021 SCM, and include numerical correlation values. Biomarkers bacterial (*i*15:0, *ai*15:0, 15:0, 15:1, *i*16:0, *ai*16:0, *i*17:0, *ai*17:0, 17:0, 17:1, and 18:1ω6), diatom (16:1ω7/16:0), flagellate ($C_{18}$PUFA/$C_{16}$PUFA), DHA/EPA (dinoflagellates) and coastal margin (18:3ω3 + 18:2ω6).

three lipid biomarkers, total lipids, was significantly higher in the average bottom depth group compared to the high dissolved oxygen/low salinity OceanMet group (S5 Table). Deviating from surface waters, PUFAs were the highest, proportionally, among the saturation groups, on average, ranging between 15.8–48.7%. The MUFAs ranged between 8.1–42.3%, and SFAs ranged between 20.1–75.9% (Fig 2). Of the 17 FA above 1%, 11 FAs (20:0, 16:1ω7, 16:4ω3,

16:4ω1, 18:1ω7, 18:2ω6, 18:3ω3, 18:4ω3, 20:5ω3, and 22:6ω3) differed significantly among OceanMet groups, and, of the three saturation sums, ΣMUFA differed significantly (Fig 2; S5 Table).

Of the eight FA biomarkers, four (DHA/EPA, diatom, flagellate, and coastal margin) differed significantly among Ocean-Met groups (S5 Table). When plotted in non-parametric space utilizing the eight FA biomarkers and the three lipid metrics, the overall distance among OceanMet groups was significant (PERMANOVA; df = 4, pseudo-F = 3.29, p = 0.001). Pairwise PERMANOVA results showed that the significant difference among all OceanMet groups was driven mainly by pairwise differences between the epipelagic OceanMet group, and all other OceanMet groups (S6 Table). In two axes, the PCoA explained 67.2% of the variability among the OceanMet groups using the eight FA biomarkers and two lipid (total lipids and TAG/PL) metrics (Fig 3).

**2019 SCM correlations.** In contrast to surface waters, the SCM showed fewer significant correlations among the 13 hydrographic and chemical properties and lipids and FAs. Of the seven oceanic measurements, five differed significantly with at least one lipid class, lipid metric, FA, or FA biomarker, mainly negatively; bottom depth and dissolved oxygen were not correlated with any lipid or FA (S4 Table). Salinity significantly correlated with the most lipid classes and lipid metrics (two of 11) and strongly, negatively correlated with TAG/PL. Of the lipid classes and lipid metrics, only three significantly correlated with at least one oceanic measurement: total lipids positively correlated with fluorescence, and ALC and TAG/PL positively correlated with salinity (Fig 4). Fluorescence, generally, negatively correlated with coastal FA and biomarker signatures, and positively correlated with diatom-associated FAs and biomarkers (13 of 18). The FA with the most correlations with the seven oceanic measurements was 18:1ω7 (three of seven), mainly positively (S4 Table).

Of the five inorganic nutrient measurements, all but nitrate significantly correlated with at least one lipid class, FA, or FA biomarker, mainly negatively (S4 Table). The lipid class with the most significant correlations was ALC (three of 14), mainly negatively, although ALC most strongly, positively, correlated with ammonia. No lipid class or metric correlated with more than one inorganic nutrient measurement (S4 Table). Phosphate had the most significant correlations of FAs (seven of 17) and FA biomarkers (five of eight); phosphate was most strongly (correlation coefficient > 0.5), negatively associated with 18:1ω7 and DHA/EPA and most strongly, positively correlated with 20:5 ω,3. Additionally, phosphate negatively correlated with ALC, and the FAs 20:0, 18:3ω3, and the Σω6, the coastal origin biomarkers, and was positively correlated with the FAs 16:1ω7, 16:4ω1, 22:1ω9, and the diatom and flagellate biomarkers. The FA with the most significant correlations with the six inorganic measurements was 18:3ω3 (two of five), all negatively, and the biomarkers that showed the most correlations were DHA/EPA and Σω6 (two of five), all negatively (Fig 4; S4 Table).

## 2021

### 2021 Sample Collection

**Surface. 2021 Surface hydrographic properties.** Between August 15 – October 4, 2021, 42 surface and 42 SCM phytoplankton samples were collected sequentially from stations in the CA (Fig 1)(S1 Table). Six oceanic and six inorganic nutrients were measured within the East Hudson Strait, Baffin Bay, and Canadian Arctic Archipelago-East stations; however, only four oceanic measurements were measured within the Canadian Arctic Archipelago and Beaufort Sea (unlike in 2019, light transmission and dissolved oxygen were not measured at those stations). Therefore, only the four common ocean measurements were used in the cluster SIMPROF (S4 Fig). The initial analysis yielded no clusters, so two separate analyses were run on each CA region: Baffin Bay and Canadian Arctic Archipelago. The cluster SIMPROF for the Baffin Bay region created six OceanMet groups, leaving one excluded station that did not fit within any group based on similar ocean metrics (S5 Fig). The cluster SIMPROF for the Canadian Arctic Archipelago region yielded no clusters (S2-C Fig). Therefore, the stations within the Canadian Arctic Archipelago region, including the Beaufort Sea and Queen Maud channels, were combined into one OceanMet group (S4 Fig; S1 Table). This combination resulted in a final count of seven OceanMet groups for 2021 surface waters: 1) low light transmission/high fluorescence, 2) average temperature, 3) large

bottom depth (oceanic), 4) average temperature & light transmission, 5) low oxygen, 6) high oxygen/low temperature, and 7) high silica/low salinity (Table 2). Of the seven resulting OceanMet groups in 2021 surface waters, one region was also defined by hydrographic properties: East Barrow Strait. The East Hudson Strait, Davis Strait, and North Water Polynya areas were included across five of the seven OceanMet clusters, including within clusters that ranged in light transmission (low *vs* average light) and oxygen saturation (low *vs* high oxygen saturation).

**Table 2. 2021 OceanMet summary.**

| | Regions within each cluster | Stations in cluster (*n*) | Shorthand area name | Description | Same region? | Significantly higher | Significantly lower | Shorthand ocean metric description |
|---|---|---|---|---|---|---|---|---|
| **Surf.** | East Hudson Strait, North Water Polynya-West | 2 | EHS/NWP | Southern EHS station, west NWP station | No | Fluorescence | Light transmission | low light/high fluorescence |
| | East Hudson Strait, Davis Strait, Davis Strait-West, Lancaster Sound | 5 | EHS/DS /LS | Northern EHS station, west and mid DS stations, and southern LS station | No | Average temperature | | average temperature |
| | Davis Strait, Davis Strait-West | 3 | DS/ DS-West | More northern DS stations | Yes | Bottom depth | | large bottom depth (oceanic) |
| | Davis Strait, Davis Strait-West, Lancaster Sound, North Water Polynya, East Barrow Strait | 12 | DS/LS/ NWP | Mostly coastal DS stations, southern LS station, and middle channel NWP station, and ESB stations | No | Average temperature, light transmission | | average temperature and light transmission |
| | North Water Polynya | 3 | NWP | All NWP stations | Yes | | Dissolved oxygen | low oxygen |
| | East Barrow Strait | 2 | EBS | All EBS stations | Yes | Dissolved oxygen | Temperature | high oxygen/low temp |
| | Queen Maud Channel, Beaufort East and West | 13 | BF/CAA | All Queen Maud Channel and Beaufort stations | No | Silica | Salinity | high silica/low salinity |
| **SCM** | East Hudson Strait, Davis Strait, Davis Strait-West, Lancaster Sound, North Water Polynya, North Water Polynya-West | 10 | EHS/NWP /LS | An equal mix of EHS, DS, NWP, and LS stations that are both offshore and coastal | No | Light transmission | Dissolved oxygen | high light/low oxygen |
| | Davis Strait, Davis Strait-West | 3 | DS/ DS-West | More northern DS stations | Yes | Bottom depth, fluorescence | | large bottom depth (oceanic)/ high fluorescence |
| | Davis Strait, North Water Polynya | 5 | DS/NWP | More southern DS stations and NWP stations | No | Average light transmission, fluorescence | | Average light transmission & fluorescence |
| | Davis Strait-West, East Barrow Strait | 8 | DS-West/ EBS | Coastal DS stations, all of EBS stations | No | Light transmission, dissolved oxygen | | high light & oxygen |
| | Queen Maud Channel, Beaufort East and West | 13 | BF/CAA | All Queen Maud Channel and Beaufort stations | No | | Salinity | low salinity |

A descriptive summary of the OceanMet groups produced by hierarchical cluster analysis (SIMPROF) based on seven ocean metrics in surface (S2-A;C, S3-A Figs) and sub-chlorophyll maximum (SCM) (S2-B;D, S3-B Fig) waters that were collected between August 15th – October 4th, 2021. Shorthand names include East Hudson Strait (EHS), North Water Polynya (NWP), Davis Strait (DS), Lancaster Sound (LS), East Barrow Strait (EBS), Beaufort Sea (BF), and Canadian Arctic Archipelago (CAA).

The hydrographic properties for the surface defined three regions: northern Davis Strait, North Water Polynya, and East Barrow Strait. Stations within these regions represented distinct areas around Baffin Bay, respectively (Fig 1). Davis Strait northern stations were defined by being more oceanic compared to the rest of the OceanMet groups. North Water Polynya was defined by having low oxygen and East Barrow Strait by high dissolved oxygen and low temperature (Table 1; S6 Fig). Of the six ocean metrics (unlike 2019, oxygen saturation is not reported) and four inorganic nutrient measurements (unlike 2019, nitrate sensor is not reported) averaged across the seven clusters, all but nitrate + nitrite and phosphate varied significantly among clusters (S6-S7 Fig). For the BF/CAA group, light transmission and dissolved oxygen measurements were not made so those calculations only reflect the average of the Baffin Bay region; ammonium was only determined in the BF/CAA.

**2021 Surface total lipids, lipid classes, and fatty acids.** Of the eight lipid classes, only ST was significantly higher in the average temperature and light transmission OceanMet group compared to all other groups, whereas one of the three lipid metrics, TAG/PL, was significantly higher in the low light/high fluorescence OceanMet group compared to all other groups (S7 Table). The SFAs were the highest, proportionally, among the saturation groups, on average, ranging between 31.7–83.9%. The MUFAs ranged between 7.1–40.7%, and PUFAs between 6.2–49.2% (Fig 5). Of the 17 FA above 1%, 11 FAs (myristic acid; 14:0, palmitic acid; 16:0, 18:0, 16:1ω7, 16:4ω1, 18:1ω7, 22:1ω9, 16:3ω3, 20:5ω3, docosapentaenoic acid, DPA, 22:5ω3, and 22:6ω3) differed significantly among OceanMet groups (S7 Table). All three saturation sums, ΣSFA, ΣMUFA, ΣPUFA, differed significantly among OceanMet groups (Fig 5; S7 Table). Generally, these differences resulted from the high silica/low salinity OceanMet group being significantly higher in ΣSFA, and therefore lower in ΣMUFA compared to most of the other OceanMet groups. The high silica/low salinity OceanMet group was only significantly lower in ΣPUFA compared to two OceanMet groups: the oceanic and average temperature/light transmission groups (S7 Table).

Of the eight FA biomarkers, four (PUFA/SFA, Σω3, bacterial, and diatom) differed significantly among OceanMet groups (S7 Table). Generally, this result parallelled that seen within the individual FAs; the high silica/low salinity OceanMet group was significantly lower in all four FA biomarkers compared to both the oceanic and average temperature/light transmission groups (S7 Table). When plotted in non-parametric space using the eight FA biomarkers and the three lipid biomarkers, the distance among OceanMet groupings was significant (PERMANOVA; df = 6, pseudo-F = 2.78, p = 0.001). Pairwise PERMANOVA results showed that the significant difference among all OceanMet groups was driven entirely by pairwise differences among the high/silica/low salinity OceanMet group, and groups that were either oceanic or were defined by average temperatures (S8 Table). In two axes, the PCoA explained 74.5% of the variability among the OceanMet groups using the eight FA biomarkers and two lipid (total lipids and TAG/PL) metrics (Fig 6).

**2021 Surface correlations.** Of the six oceanic measurements, all significantly correlated with at least one lipid class, lipid metric, FA, or FA biomarker, mainly positively (S9 Table). Lipid classes and metrics correlations were limited; only fluorescence positively correlated with TAG/PL. Salinity significantly correlated with the most individual FAs (11 of 17), mainly positively, while temperature significantly correlated with the most FA biomarkers (four of eight), all positively, just like in 2019 surface waters (S9 Table). Salinity strongly correlated positively with 14:0, and temperature strongly correlated positively with 18:3ω3 and 18:4ω3. Most of the individual FAs significantly correlated with two of the 18-carbon FA metrics, mainly bottom depth and salinity; 14:0, 16:1ω7, MUFA, 16:3ω3, 16:4ω1, and 20:5ω3 were positively correlated, and 18:0, ΣSFA, and 22:5ω3 were negatively correlated (Fig 4). The FA biomarker with the most correlations with oceanic measurements was PUFA/SFA, positively correlating with bottom depth, temperature, and salinity (S9 Table).

Of the four inorganic nutrient measurements, all but nitrate were significantly correlated with at least one lipid class, lipid metric, FA, or FA biomarker, mainly positively (S9 Table). Ammonium, which was only determined in the BF/CAA OceanMet group, strongly correlated positively with lipid classes TAG and AMPL, and the lipid metric TAG/PL, while also strongly correlating negatively with phosphate. Silica correlated with both the most FAs (11 of 17), mainly negatively, and the most FA biomarkers (four of eight), all negatively. Silica strongly, negatively correlated with 14:0 and ΣMUFA, while correlating strongly, positively with 18:0 and ΣSFA. No FA correlated with more than one nutrient. The bacterial FA

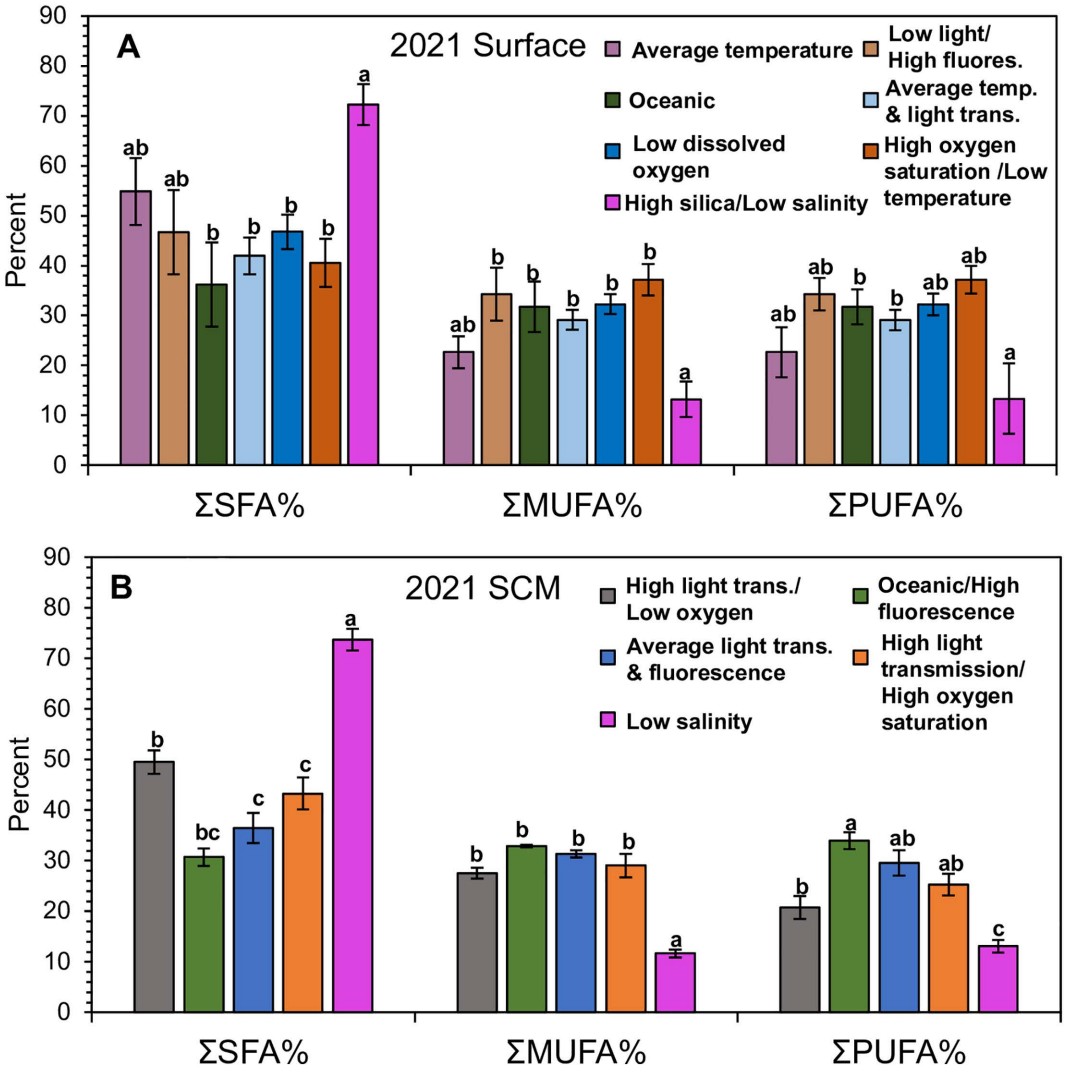

**Fig 5. 2021 Phytoplankton fatty acid summary.** Summary of the phytoplankton fatty acid sums and biomarkers, gathered from surface (A) and subsurface-chlorophyll maximum (SCM) (B) waters from August 15 – October 4, 2021. Fatty acid sums are grouped by OceanMet group and averaged (±SE) across saturated fatty acids (SFA%; no sig. differences), monounsaturated fatty acids (MUFA%), and polyunsaturated fatty acids (PUFA%; no sig. differences). Tukey comparisons (lowercase letters) indicate significant (p ≤ 0.05) differences among groups.

biomarker correlated with the most inorganic nutrients (two of four), correlating negatively with silica and positively with ammonium (Fig 4; S9 Table).

### SCM

**2021 SCM hydrographic properties.** Unlike the surface waters analyses, the initial SIMPROF analysis yielded three clusters across the entire CA (S4 Fig), leaving one excluded station. However, because the replicate number within each group was extremely uneven (two of the groups contained less than four stations), the decision was made to run two separate analyses on each CA region, identical to that for surface waters analyses. The cluster SIMPROF for the Baffin Bay region created four OceanMet groups, leaving two excluded stations that did not fit within any group based

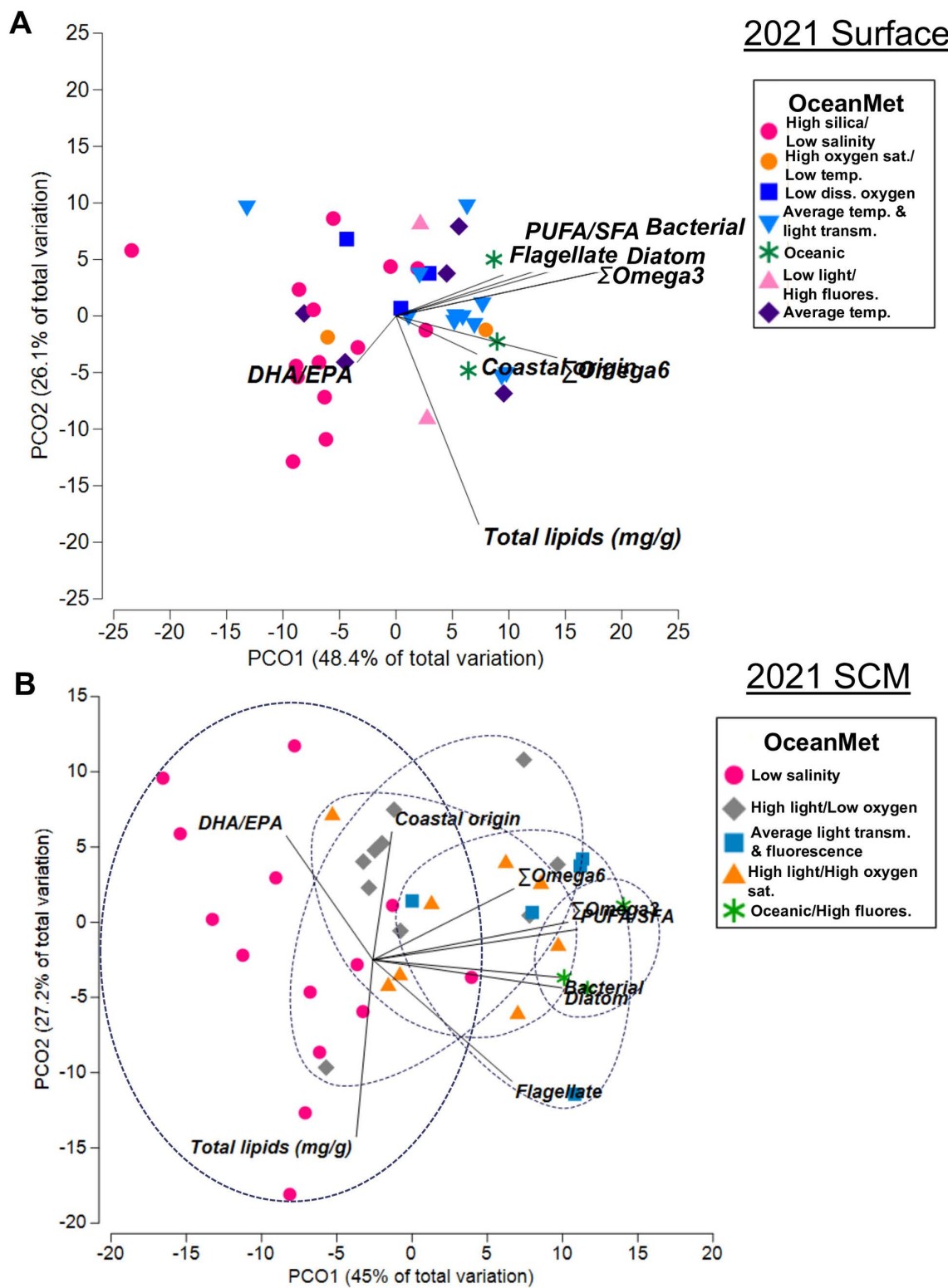

**Fig 6. 2021 PCoA plot.** Fatty acid and lipid biomarkers plotted on a principal coordinate analysis (PCoA) plot, and grouped by OceanMet (dotted blue lines) for surface (A) and subsurface-chlorophyll maximum (SCM) (B) waters from August 15 – October 4, 2021. OceanMet groups are also listed by shorthand area name (Table 2). Biomarkers include total lipids (mg/g), PUFA/SFA, ω3, ω6, bacterial ($i$15:0, $ai$15:0, 15:0, 15:1, $i$16:0, $ai$16:0, $i$17:0, $ai$17:0, 17:0, 17:1, and 18:1ω6), diatom (16:1ω7/16:0), flagellate ($C_{18}$PUFA/$C_{16}$PUFA), DHA/EPA (dinoflagellates) and coastal margin (18:3ω3 + 18:2ω6).

on similar ocean metrics (S4 Fig). The cluster SIMPROF for the Canadian Arctic Archipelago region yielded no clusters. Therefore, the stations within the Canadian Arctic Archipelago region were combined into one OceanMet group (S4 Fig; S1 Table). This combination resulted in a final count of five OceanMet groups for 2021 SCM waters: 1) high light/low oxygen, 2) high bottom depth (oceanic)/high fluorescence, 3) average fluorescence, 4) high light/high oxygen, and 5) low salinity (S5 Fig; Table 2 and S1 Table). No areas were singularly represented in any OceanMet group in 2021 SCM waters. However, varying levels of both light transmission (average *vs* high) and oxygen saturation (low *vs* high) were represented in the OceanMet clusters. In the SCM, one region was also defined by hydrographic properties: northern Davis Strait. As in surface waters, northern David Strait stations were defined by being more oceanic and having a higher fluorescence compared to the rest of the OceanMet groups (Table 1; S6 Fig). Of the six ocean metrics (unlike 2019, oxygen saturation is not reported) and four inorganic nutrient measurements (unlike 2019, nitrate sensor is not reported) averaged across the five clusters, all but temperature, nitrate + nitrite, silicate, and phosphate varied significantly among clusters (S6-S7 Fig).

**2021 SCM total lipids, lipid classes, and fatty acids.** Of the eight lipid classes, only FFA was significantly different among OceanMet groups, and there were no significant differences for any lipid class or metric (S10 Table). The SFAs were the highest, proportionally, among the saturation groups, on average, ranging between 28.3–82.6%. The MUFAs ranged between 8.2–41.0%, and PUFAs between 8.5–40.3% (Fig 5). Of the 17 FA above 1%, 12 FAs (14:0, 16:0, 18:0, 16:1ω7, oleic acid (18:1ω9), 18:1ω7, 22:1ω9, 16:3ω3, 16:4ω1, 20:5ω3, 22:5ω3, and 22:6ω3) differed significantly among OceanMet groups, deviating only slightly from the 2021 surface waters with the addition of 18:1ω9. All three saturation sums, ΣSFA, ΣMUFA, and ΣPUFA, differed significantly among OceanMet groups (S10 Table). Generally, as in the 2021 surface waters, these differences resulted from the high silica/low salinity OceanMet group being significantly higher in ΣSFA, and therefore lower in ΣMUFA compared to most of the other OceanMet groups; however, in the SCM waters, this pattern also translated to the ΣPUFA sum and PUFA FAs in general (S10 Table).

Of the eight FA biomarkers, five (PUFA/SFA, Σω3, bacterial, diatom, and flagellate) differed significantly among OceanMet groups (S10 Table), similar to the 2021 surface waters. Generally, these differences resulted from the same pattern seen within the individual FAs; the low salinity OceanMet group was significantly lower compared to the other OceanMet groups in all but one comparison; the low salinity OceanMet group was only significantly lower than the oceanic/high fluorescence OceanMet group for the flagellate biomarker (S10 Table). Unlike in 2019 and 2021 surface waters where no significance was found among groups, the proportions of EPA + DHA varied significantly in that the low salinity OceanMet group was lower than the majority of the other groups (df = 4, F = 23.83, p < 0.001) (S10 Table).

When plotted in non-parametric space utilizing the eight FA biomarkers and the three lipid biomarkers, the distance among OceanMet groupings, was significant (PERMANOVA; df = 4, pseudo-F = 3.92, p = 0.001). Pairwise PERMANOVA results showed that the significant difference among all OceanMet groups was driven entirely by pairwise differences among the low salinity OceanMet group, and all other OceanMet groups (S11 Table). In two axes, the PCoA explained 72.2% of the variability among the OceanMet groups using the eight FA biomarkers and two lipid (total lipids and TAG/PL) metrics (Fig 6).

**2021 SCM correlations.** Of the six oceanic measurements, all but dissolved oxygen significantly correlated with at least one lipid, lipid metric, FA, or FA biomarker, mainly positively (S9 Table). Light transmission significantly correlated with the most lipid classes or metrics (four of 11), mainly negatively, and strongly correlated negatively with FFA. The lipid class that correlated with the most oceanic measurements was PL (three of six), mainly negatively. Light transmission also significantly correlated with the most individual FAs (ten of 17), mainly negatively, and FA biomarkers (five of eight), mainly negatively (Fig 4; S9 Table). There were three individual FAs and one FA biomarker that correlated with five of the six oceanic measurements: 16:1ω7, 16:4ω1, 16:3ω3, and the diatom biomarker. While the significant correlation strength varied, all followed the same pattern; there was a positive correlation for bottom depth, salinity and fluorescence and a negative correlation with temperature and light transmission (S9 Table).

All four inorganic nutrient measurements were significantly correlated with at least one lipid, lipid metric, FA, or FA biomarker, mainly negatively in the 2021 SCM waters (S9 Table). No inorganic nutrient correlated with any lipid class or metric. Phosphate strongly correlated negatively with 18:1ω9, 18:3ω3, 22:6ω3, DHA/EPA, and the coastal margin biomarker (Fig 4; S9 Table).

## Discussion

The first aim of our investigation was to assess which CA groups were significantly different based on lipid metrics and FA biomarkers. By "groups", we are referring to the OceanMet cluster groups, created based on similar ocean metrics, as opposed to running our analyses on previously named Arctic regions. This was done to assess if any variation of lipid metrics and FA biomarkers by region was a result of hydrographic properties; by clustering regions based on similar environments, we control for region/latitude. Of course, regions tend to have similar ocean environment profiles, and we did indeed find that Arctic regions tended to have similar ocean metrics, but this was not always the case. Our study highlights that significant differences are the result of environmental differences, and not always necessarily of the region in the CA. Indeed, multiple metrics (oxygen saturation, light transmission, fluorescence) varied across OceanMet groups that included the same areas, across both years and both surface and SCM waters, giving us the opportunity to control for latitude. Given the uncertainty in estimating future phytoplankton FA profiles in the Arctic Ocean, the need for combination analyses on environmental conditions, phytoplankton assemblages, and phytoplankton energetics is increasing across the Arctic. These results are meant to be complementary to the previously published studies focusing on energetics in various regions of the CA, and Arctic at large [29,33,65–68].

To add perspective to our aim, we correlated the lipid metrics and FA biomarkers for all available hydrographic properties within each year, providing phytoplankton lipid and FA profiles across two years, spanning multiple CA regions. Ideally, the years would be comparative, but there was too much variation; however, we found the variation in region/latitude was a strength. In 2019, we were able to focus on a more confined location, Baffin Bay, whereas in 2021, our view expanded to include a large component of the entire CA, spanning from the Baffin Bay across to the Beaufort Sea. If patterns emerged across both years, it would be indicative of an important hydrographic property mediating phytoplankton energetics, regardless of the region.

### Phytoplankton condition

Polar lipids, both as its own lipid class (encompassing both AMPL and PL) and as a part of the TAG/PL metric, was an important condition marker that correlated with hydrographic properties. Reduced polar lipids characterized OceanMet groups that were generally clustered based on high fluorescence, high temperatures, low salinity, and low inorganic nutrients (representative of coastal/river-runoff conditions). In 2019, this pattern was more pronounced in that the clustering tended to capture regions within Baffin Bay. Variation in phytoplankton bloom phenology for the Baffin Bay region can differ based on proximity to either Greenland or Ellesmere Island because of currents (Fig 1) [68]. The positive correlation between temperature and the polar lipid class, and a negative correlation with dissolved oxygen corroborated this interpretation in the surface waters (S4 Table). This result is consistent with the warmer waters of the West Greenland current in comparison to the cold Baffin Island Current [69]. Similarly, the 2019 SCM OceanMet lipid group differences were largely driven by high proportions of the polar lipid class. The apparent polar lipid class domination in the Greenland side group, along with the most shallow and coastal North Water Polynya stations, suggests either a cellular membrane modification or increased energy usage and subsequent depletion, proportionally, of TAG. TAG/PL was significantly higher in OceanMet groups clustered due to high fluorescence across the CA (2021) as well.

TAG and total lipids varied significantly, although this was only seen in the 2019 SCM dataset (Fig 2B). TAG was proportionally higher among the coastal OceanMet clusters that were more coastal, and with lower nutrients. Scarce

inorganic nutrients, especially nitrogen (i.e., nitrate, nitrite, or ammonia), typically lead to depleted polar lipids [61,70]. Indeed, total lipids were significantly higher in those more average OceanMet groups.

The high temperature/low oxygen saturation, nitrate sensor, phosphate (a cluster composed of the Store Hellefiske Bank stations) and epipelagic (a cluster composed of Lancaster Sound, Nares Strait, and North Water Polynya stations) OceanMet groups were sampled within a three week window, so it is possible data are showing a decrease in primary productivity as the summer progresses, i.e., revealing the lag in bloom timing. This scenario was also likely seen in the 2021 dataset given that the clusters containing high fluorescence also contained stations that were sampled 2–3 weeks a part (East Hudson Strait compared to North Water Polynya in 2021).

Nitrogen is a limiting nutrient in the Arctic [71–73] and has been shown to be related to phytoplankton cell division and lipid class composition [42]. When limited, nitrogen should induce, or positively correlate, with polar lipids. In our dataset, nitrate did not positively correlate with AMPL or PL in either the surface or SCM waters, suggesting phytoplankton in 2019 in these areas were either nitrogen limited, or that competing variables masked the influence of nitrogen. Some common and ecologically important Arctic diatoms may indeed be able to adapt to varying nitrogen concentrations [74].

Relative to the other groups, higher temperatures characterized OceanMet clusters with Store Hellefiske Bank having the highest temperatures in both the surface and SCM waters and also represented some of the shallowest stations sampled in 2019 (S4 Fig). In surface waters, bottom depth, temperature, and light transmission were positively correlated with phytoplankton ΣPUFA and Σω3; this relationship did not translate into the OceanMet clusters, which varied significantly for either of these two FA condition biomarkers (S4 Table). This pattern was identical for 2021 SCM waters, except that the pattern now extended across the archipelago; OceanMet groups containing average temperatures were significantly lower in ΣPUFA compared to the oceanic OceanMet group. The average temperature OceanMet groups in 2021 were shallower, warmer, with less light transmission, and with less fluorescence compared to the oceanic OceanMet groups of 2019. Phytoplankton, as with many other organisms, adapt to changing temperatures by modifying cell membrane structure, a process called homeoviscous adaption [75]. To adapt to warming, cell membranes increase SFA content at the expense of PUFA content in order to maintain cell membrane rigidity [14,30]. These FA patterns are consistent with the low salinity/low silica OceanMet group (Beaufort Sea and Canadian Arctic Archipelago stations) is relatively warmer compared to the other OceanMet groups. Similarly, the high light/low oxygen OceanMet group had similar characteristics and thus, positive correlations with the low salinity/low silica group. The high light/low oxygen OceanMet group was also warmer and lower in ΣPUFA, aligning with previous studies focusing on increasing temperatures and the resulting ΣPUFA profiles [27,76]. In both 2021 surface and SCM waters, the FA condition biomarkers, PUFA/SFA and Σω3 were significantly higher in the oceanic OceanMet groups (i.e., the stations representing Baffin Bay in 2021) compared to the low salinity OceanMet group (S7 Table).

In general, 2021 surface and SCM water patterns were similar (Fig 6; S9 Table). ΣSFA was negatively correlated with bottom depth and salinity, which algins with proportionally higher FAs in the low salinity OceanMet group. ΣMUFA was positively correlated with bottom depth and salinity, whereas ΣPUFA was positively correlated with temperature (S9 Table). The PUFA/SFA and Σω3 FA condition biomarkers were positively correlated with temperature as well, whereas PUFA/SFA was also correlated with bottom depth and salinity (S9 Table). This positive correlation with temperature and Σω3 in our dataset contradicts results seen in Hixon and Arts (2016), but this difference may result from their study investigating a wide range of phytoplankton groups from both freshwater and marine sources; notably, EPA and DHA, and EPA + DHA, rarely correlated with temperature in either year. In 2021 surface waters, the high silica/low salinity OceanMet group included a significantly higher proportion of SFAs compared to the groups that were colder and more oxygenated groups (and also in Baffin Bay mostly). This was driven by proportionally higher 16:0 and 18:0 than 14:0. The opposite pattern seen in ΣSFA was, by extension, evident for both ΣMUFA and ΣPUFA, although the high silica/low salinity group differed significantly from the oceanic OceanMet group for ΣPUFA (S9 Table). While stark differences were observed between the more eastern and western CA (more eastern was significantly higher in SFAs), we were unable to ascertain whether this

difference solely related to latitude (or time) or the OceanMet group difference; no other OceanMet groups were defined by low silica and high salinity values.

The Σω3 condition biomarker, however, reveals that the metric EPA+DHA never correlated with temperature in any year, and only correlated positively with light, depth, and phosphate, and negatively with silica and nitrate. EPA and DHA are often considered together, especially in fish research, yet our research does not support the significant influence of cold environments on EPA and DHA content in fish [77]. In 2019, DHA alone and DHA/EPA was significantly higher in the "high temperature" groups. Indeed, DHA and DHA/EPA positively correlated with temperature increases across years and water depths, and EPA correlated with a temperature decrease (2019 SCM), suggesting temperature increases may affect specific FAs within FA groups such as SFA, MUFA, and PUFA. In regard to climate change, assessing Σω3 levels in phytoplankton as temperatures increase may require more nuanced consideration of different FAs, especially EPA and DHA, as opposed to generalizing PUFA and Σω3 [78].

## Phytoplankton assemblages

Across both years and water column depths, phytoplankton FA composition mostly correlated with oceanic measurements, especially salinity, temperature, and light transmission, as opposed to the inorganic nutrient measurements; the notable exceptions were phosphate and silica, and these two inorganic nutrient patterns became more important with the wider geographic scope in 2021. The FA assemblage diatom and coastal margin biomarkers that were more important, i.e., correlated with the same hydrographic properties that were also responsible for the resulting OceanMet groupings, were the diatom and coastal margin biomarkers; the diatom biomarker was higher in OceanMet groups that tended to be epipelagic, have high fluorescence and high salinity, whereas the coastal margin biomarker was higher in OceanMet groups with higher temperatures, lower nitrate sensor values, and lower phosphate. Given the timing of these samplings, the pattern reflects the progressive transition of the phytoplankton assemblage from diatoms in the spring, to a more flagellate and dinoflagellate dominated community; this would appear first in the more southern regions [79].

Constituting a significant portion of the FA pool (2–5%), the coastal-margin biomarker (i.e., vascular plant and macroalgal FA), might be higher in the high temperature clusters because of closer proximity to land (both in 2019 and 2021), and immediate influx of advected Atlantic waters which are significantly warmer and vary in nutrient composition compared to the more polar waters [80–82] (S4 Fig). Vascular plant and green macroalgal inputs possibly accumulate in the warmer surface waters and not being detected beyond the SCM. This difference was particularly evident in 2021, as the sampling area covered much more of the Canadian Arctic Archipelago compared to 2019. The greater presence of the coastal margin biomarker in the high silica/low salinity OceanMet group therefore likely reflects closer proximity of stations to the Mackenzie River (Fig 1). The importance of coastal influences on Arctic primary production [48] aligns with the more discernible phytoplankton assemblage in this region, and by extension, its unique hydrographic properties (i.e., high silica and low salinity). Indeed, the low diatom markers in this OceanMet group relative to the others suggest that while nutrient input in coastal areas is high, suspended matter from the terrestrial riverine output may impede growth due to limited light penetration [83].

The OceanMet group with average bottom depth and fluorescence (all East Hudson Strait stations in 2021) was differentiated by bacterial and DHA/EPA biomarker ratios, together suggesting a phytoplankton community dominated by dinoflagellates and haptophytes. Although DHA/EPA, and by extension DHA on its own, has been used as a dinoflagellate marker [55], other studies report that these two do not correlate well in this Arctic system [29] and may result from an influx of cryptophytes [84]. A consistent pattern throughout is that EHS, in both the surface and SCM waters, was rarely differentiated by any group in either 2019 or 2021; this OceanMet group (and region) consistently exhibited lipid and FA values in-between the profiles exhibited by the more "southern" Baffin Bay and "northern" Baffin Bay OceanMet groups. The poleward expansion of temperate phytoplankton species [85] could also explain the more dinoflagellate dominated population, especially given the expansion of the North Atlantic Current [86]. In 2019 especially, the DHA/EPA significance

in both the surface and SCM waters resulted from a difference between high temperature/low nutrient and more average fluorescence OceanMet groups, where the high temperature/low nutrient ratio was larger. Overall, it appears that the 2019 surface and SCM FA assemblage biomarkers suggest a phytoplankton community consisting of dinoflagellates more so than diatoms and flagellates (these clusters all also happen to have Store Hellefiske Bank stations). Because both higher temperature and lower oxygen saturation define these clusters, the results indicate an environment more suitable for dinoflagellate growth. Further, spring blooms dominated by dinoflagellates have been reported in the Arctic [87]. However, based on when these samples were collected for this study (July-October in both years), the surface water biomarkers may be capturing the more dinoflagellate-rich community in the south of Baffin Bay [88].

DHA/EPA and the bacterial biomarkers were positively correlated with temperature and salinity and negatively correlated with dissolved oxygen and oxygen saturation (Fig 4), indicating warmer temperatures are correlated with higher DHA and ΣPUFA. These results contradict published studies that have found ΣPUFA in cultured phytoplankton increase proportionally in response to cold temperatures in order to maintain membrane fluidity [27,76]. Further, the results from our study support published studies that have reported DHA decreases in the Central Arctic Ocean [66]. Flagellates showed the complete opposite trend and were negatively correlated with temperature and salinity, and positively with dissolved oxygen and oxygen saturation (Fig 4). Oxygen saturation can be used as an indicator of the primary productivity profile that was present in the days or weeks prior to sampling, and could affect lipid composition in stressed organisms with potential inflammation leading to subsequent oxidation [89–91]. However, dissolved oxygen and oxygen saturation are much less likely to affect lipid values directly in healthy phytoplankton communities; lack of dissolved oxygen in the marine environment can limit primary production given its close coupling to the biogeochemical cycle of both nitrogen and phosphate [92], and low oxygen levels would affect phytoplankton population size instead of lipid physiology.

Nitrogen and phosphate showed identical trends in surface waters in that they strongly, negatively correlated with DHA/EPA, bacterial, and diatom, and strongly positively correlated with the flagellate and coastal origin biomarkers (S4, S9 Tables), as reported in Marmillot et al. 2020 [29]. Nitrogen was sometimes negatively and positively correlated with both diatom and dinoflagellate markers in the surface. Although abundance or net primary productivity was not measured, these results suggest nitrogen limitation might affect some phytoplankton assemblages more than others. The diatom biomarker was positively correlated with fluorescence, oxygen saturation, phosphate, and negatively with silica. Given that phosphate is used as an indicator of phytoplankton succession throughout the season [93], the lower values in the lower latitude dominated OceanMet clusters makes sense, especially given the low diatom biomarker values for those groups. However, the negative correlation with silica and lower values of the diatom marker in the high silica OceanMet group was surprising given their reliance on this nutrient [94]. As previously discussed, this pattern may again reflect a timing issue in that our sampling did not capture the diatom bloom [95]. The low salinity that also defined the high silica OceanMet group could have been a contributing factor, given a significant positive salinity correlation; this would be suggestive of regions in the CA more likely populated with flagellates [43]. Alternately, low diatom biomarker values may reflect zooplankton consumption or sampling season.

## Conclusions

Phytoplankton biomass is expected to increase with climate change in the Arctic [96,97]. However, there is uncertainty regarding the availability of Σω3 in future Arctic marine ecosystems, even with this expected phytoplankton biomass increase. Estimates of primary Σω3 must consider the combination of environmental drivers, the variation in phytoplankton taxa, the estimated contribution of nutrients from land and melting sea ice, and the estimation of zooplankton grazing rates. In our study, we clustered stations by ocean metrics instead of comparing Arctic regions, providing a more nuanced view of phytoplankton lipids in this region. Across two years and both surface and SCM, we generally found that clusters defined by being oceanic, with higher salinity, higher phosphate, and higher oxygen saturation were more likely to contain higher proportions, and positively correlate with diatom markers, PUFA/SFA, Σω3, and lower amounts of $C_{18}$ PUFAs. In a

warmer, ice-free Arctic, these ocean metrics will be important factors in mediating energy availability at the bottom of the food web.

No one hydrographic property or nutrient could predict Σω3 or ΣPUFA consistently, or even regionally in 2019 or 2021. However, Σω3% correlated positively with temperature in both years in the surface waters, as did DHA (but notably not EPA). Indeed, EPA never correlated with temperature, and EPA+DHA% rarely correlated with temperature. In 2021, temperature correlated strongly with other ω3 FAs, including ALA and 18:3ω4 in both the surface and SCM, indicating that the while EPA (diatom and coastal macroalgae indicator), negatively correlated with temperature, other FAs contributing to the ω3 pool may increase with an increase in temperature in certain areas of the CA. Indeed, we did find that certain areas of the CA could be differentiated based on the lipid profiles, as many clusters inevitably grouped together based on both location and ocean metrics.

Higher diatom markers, PUFA/SFA, and Σω3% tended to co-occur in clusters defined by stations that were more oceanic and in the eastern CA (Baffin Bay and North Water Polynya) as opposed to the Beaufort Sea and the western Canadian Arctic Archipelago. Additionally, the condition metric, TAG:PL, correlated negatively with a multitude of hydrographic properties including temperature, salinity, light transmission, and silica. The Beaufort Sea groups were mostly defined as more shallow, less saline, and lower in fluorescence than the other simulated groups, consistent with the absence of a relatively high condition metric in phytoplankton in the Beaufort Sea and western Canadian Arctic Archipelago regions. In 2021 especially, the clusters with significantly higher condition biomarkers tended to occur in the oceanic zones within both the southern Baffin Bay area, and the coastal zones within the North Water Polynya area. Both of these areas are likely regions of productivity as suggested by their Σω3 and PUFA profiles. Initially, warmer temperatures are predicted to increase primary production, as well as zooplankton biomass [13,44]. However, whether the increase in phytoplankton biomass leads to an increase in EFAs that can sustain consumers higher in the food web is less understood.

While it is difficult to ascertain how the compounded effects of multiple climate change variables will impact the FA profiles of Arctic phytoplankton communities [11,66,98], our study provides an estimate of which ocean metrics are most important, in part, in influencing phytoplankton lipid profiles. In the CA, regions defined by lower salinities and proximity to shore were less enriched in phytoplankton EFAs than regions deeper and that have relatively higher amounts of phosphate. These regions were located in the eastern CA; direct effects of climate change on the more productive regions of Baffin Bay and the North Water Polynya may cause food web changes more quickly in those areas than changes to the Beaufort Sea or the more western Canadian Arctic Archipelago.

Phytoplankton FA proportions in our dataset were likely being largely driven by proximity to the shore, as indicated by bottom depth, fluorescence, and salinity, and the coastal margin FA biomarker. Epipelagic water characteristics thus favored diatom fatty acids such as 16:1ω7, 16:1ω4, and EPA over the saturates, which were significantly higher in the more neritic waters. The results imply that although regional differences and latitudinal patterns are important, the overarching mediator of phytoplankton FA profiles is in large part related to hydrographic properties in the Arctic. Overall, our results provide a hydrography and water chemistry-based approach to indicate areas of high phytoplankton energetic condition, irrespective of latitude.

## Supporting information

**S1 Dataset.** An overall dataset with the raw data within this manuscript.
(XLSX)

**S1 Fig. 2019 SIMPROF results.** Hierarchical cluster analysis using group averages and similarity profile permutation tests (SIMPROF; red lines) based on seven oceanic metrics (bottom depth, temperature, salinity, light transmission, fluorescence, oxygen saturation, and dissolved oxygen) applied to samples gathered from surface waters (A) and the sub-surface chlorophyll maximum (SCM)(B) from July 8th – September 3rd, 2019. Station names and subsequent cluster

grouping are termed OceanMet groups. Shorthand names include East Hudson Strait (EHS), Store Hellefiske Bank (SHB), North Water Polynya (NWP), Davis Strait (DS), Nares Strait (NS), Lancaster Sound (LS), East Barrow Strait (EBS), and Talbot Trough (TT).
(PDF)

**S2 Fig. 2019 Ocean metric averages.** A summary of the seven ocean metrics averaged (±SE) across the OceanMet groups for both surface (white) and sub-surface chlorophyll maximum (grey) gathered from July 8th – September 3rd, 2019. Letters differentiate significantly different groups (ANOVA; Tukey, $p < 0.5$) and the horizontal bar represents the overall average of both surface and the sub-chlorophyll maximum. The OceanMet groups are named by location short-hand instead of ocean metric description; refer to Table 2 for shorthand ocean metric description which includes nutrient information in OceanMet group. Shorthand names include East Hudson Strait (EHS), Store Hellefiske Bank (SHB), North Water Polynya (NWP), Davis Strait (DS), Nares Strait (NS), Lancaster Sound (LS), East Barrow Strait (EBS), and Talbot Trough (TT).
(PDF)

**S3 Fig. 2019 nutrient averages.** A summary of the six ocean nutrients averaged (±SE) across the OceanMet groups for both surface (white) and the sub-chlorophyll maximum (grey) gathered from July 8th – September 3rd, 2019. Letters differentiate significantly different groups (ANOVA; Tukey, $p < 0.5$) and the horizontal bar represents the overall average of both surface and the sub-surface chlorophyll maximum; a Kruskal Wallis test was performed for nitrate + nitrite surface, and the groups responsible for the difference are italicized. A blank column indicates variable was not collected. The OceanMet groups are named by location shorthand instead of ocean metric description; refer to Table 2 for shorthand ocean metric description which includes nutrient information in OceanMet group. Shorthand names include East Hudson Strait (EHS), Store Hellefiske Bank (SHB), North Water Polynya (NWP), Davis Strait (DS), Nares Strait (NS), Lancaster Sound (LS), East Barrow Strait (EBS), and Talbot Trough (TT).
(PDF)

**S4 Fig. 2021 SIMPROF results within the Beaufort Sea and Canadian Arctic Archipelago.** Hierarchical cluster analysis using group averages and similarity profile permutation tests (SIMPROF; red lines) based on six oceanic metrics (bottom depth, temperature, salinity, light transmission, fluorescence, and dissolved oxygen) applied to samples gathered from surface waters (A) and sub-surface chlorophyll maximum (SCM)(B) between August 15 – October 3, 2021. Non-significant clusters for both surface and SCM resulted in a division between Baffin Bay and Canadian Arctic Archipelago (CAA), and subsequent cluster analysis were done separately on each region. Based on four oceanic metrics (bottom depth, temperature, salinity, and fluorescence) clusters were not significant for CAA surface and SCM stations, resulting in a single OceanMet group.
(PDF)

**S5 Fig. 2021 SIMPROF results.** Hierarchical cluster analysis using group averages and similarity profile permutation tests (SIMPROF; red lines) based on six oceanic metrics (bottom depth, temperature, salinity, light transmission, fluorescence, and dissolved oxygen) applied to samples gathered from surface waters (A) and the sub-surface chlorophyll maximum (SCM)(B) from August 15 – October 3, 2021 in Baffin Bay and the east side of the Canadian Arctic Archipelago (CAA). Station names and subsequent cluster grouping are termed OceanMet groups. Shorthand names include East Hudson Strait (EHS), North Water Polynya (NWP), Davis Strait (DS), Lancaster Sound (LS), East Barrow Strait (EBS), Beaufort Sea (BF), and Canadian Arctic Archipelago (CAA).
(PDF)

**S6 Fig. 2021 Ocean metric averages.** A summary of the six ocean metrics averaged (±SE) across the OceanMet groups for both surface (white striped) and sub-chlorophyll maximum (grey striped) gathered between August 15 – October 3,

2021. Letters differentiate significantly different groups (ANOVA; Tukey, p < 0.05) and the horizontal bar represents the overall average of both surface and sub-chlorophyll maximum. A blank column indicates variable was not collected. The OceanMet groups are named by location shorthand instead of ocean metric description; refer to Table 2 for shorthand ocean metric description which includes nutrient information in OceanMet group. Shorthand names include East Hudson Strait (EHS), North Water Polynya (NWP), Davis Strait (DS), Lancaster Sound (LS), East Barrow Strait (EBS), Beaufort Sea (BF), and Canadian Arctic Archipelago (CAA).
(PDF)

**S7 Fig. 2021 Nutrient averages.** A summary of the five ocean nutrients averaged (±SE) across the OceanMet groups for both surface (white striped) and sub-surface chlorophyll maximum (grey striped) gathered from August 15 – October 3, 2021. Letters differentiate significantly different groups (ANOVA; Tukey, p < 0.05) and the horizontal bar represents the overall average of both surface and sub-surface chlorophyll maximum. A blank column indicates variable was not collected. The OceanMet groups are named by location shorthand instead of ocean metric description; refer to Table 2 for shorthand ocean metric description which includes nutrient information in OceanMet group. Shorthand names include East Hudson Strait (EHS), North Water Polynya (NWP), Davis Strait (DS), Lancaster Sound (LS), East Barrow Strait (EBS), Beaufort Sea (BF), and Canadian Arctic Archipelago (CAA).
(PDF)

**S1 Table. Station locations.** Station locations from which both surface and sub-surface chlorophyll maximum (SCM) phytoplankton samples were collected in both 2019 and 2021. Stations are ordered by Arctic region and the most southern latitude. OceanMet groups (Figures S1-3) are also listed. Shorthand names include East Hudson Strait (EHS), Store Hellefiske Bank (SHB), North Water Polynya (NWP), Davis Strait (DS), Nares Strait (NS), Lancaster Sound (LS), East Barrow Strait (EBS), Talbot Trough (TT), Beaufort Sea (BF), and Canadian Arctic Archipelago (CAA). *The three filters were processed for lipids and fatty acids separately, and the values were later combined.
(PDF)

**S2 Table. 2019 Surface lipid and fatty acid cluster averages.** Summary averages (±SE) of the nine phytoplankton OceanMet group lipid class percentages, lipid metrics, fatty acid percentages (>1%), and fatty acid biomarkers gathered from surface waters from July 8th – September 3rd, 2019. If significance (p ≤ 0.05) was found among groups based on a lipid or fatty acid, Tukey comparisons in the form of lowercase letters are displayed. Some fatty acids were not detected (ND). Shorthand names include East Hudson Strait (EHS), Store Hellefiske Bank (SHB), North Water Polynya (NWP), Davis Strait (DS), Nares Strait (NS), Lancaster Sound (LS), East Barrow Strait (EBS), and Talbot Trough (TT).
(PDF)

**S3 Table. 2019 Surface PERMANOVA results.** Permutational analysis of variance (PERMANOVA) pairwise results among the nine OceanMet groups (shorthand area name) created from phytoplankton gathered from surface waters from July 8th – September 3rd, 2019. Significance (p ≤ 0.05) between pairs is denoted by an asterisk next to the group. Shorthand names include East Hudson Strait (EHS), Store Hellefiske Bank (SHB), North Water Polynya (NWP), Davis Strait (DS), Nares Strait (NS), Lancaster Sound (LS), East Barrow Strait (EBS), and Talbot Trough (TT).
(PDF)

**S4 Table. 2019 Correlations.** Regression summary correlating the oceanic measurements (bottom depth, temperature, salinity, light transmission, dissolved oxygen, oxygen saturation, nitrate sensor, nitrate + nitrite, silicate, phosphate, and ammonium) with lipid class percentages, lipid biomarkers, fatty acid percentages, and fatty acid biomarkers gathered from surface (A) and subsurface chlorophyll maximum (SCM)(B) waters between July 10 – September 3, 2019 from the East Hudson Strait, through Baffin Bay, and ending in the east part of the Canadian Arctic Archipelago. Lipid and fatty

acid classes presented are above, on average amongst all samples, 1% of the phytoplankton gathered from all locations in both 2019 and 2021. If a correlation was significant (p ≤ 0.05), the correlation coefficient was highlighted as follows: 1–0.5 = orange, 0.49–0 = yellow, −0.01 – −0.49 = light green, −0.5 – −1 = green.
(PDF)

**S5 Table. 2019 SCM lipid and fatty acid cluster averages.** Summary averages (±SE) of the five phytoplankton Ocean-Met group lipid class percentages, lipid metrics, fatty acid percentages (>1%), and fatty acid biomarkers gathered from sub-surface chlorophyll maximum (SCM) waters from July 8th – September 3rd, 2019. If significance (p ≤ 0.05) was found among groups based on a lipid or fatty acid (Table S1), Tukey comparisons in the form of lowercase letters are displayed. Some fatty acids were not detected (ND). Shorthand names include East Hudson Strait (EHS), Store Hellefiske Bank (SHB), North Water Polynya (NWP), Davis Strait (DS), Nares Strait (NS), Lancaster Sound (LS), East Barrow Strait (EBS), and Talbot Trough (TT).
(PDF)

**S6 Table. 2019 SCM PERMANOVA results.** Permutational analysis of variance (PERMANOVA) pairwise results among the nine OceanMet groups (shorthand area name) created from phytoplankton gathered from sub-surface chlorophyll maximum (SCM) waters from July 8th – September 3rd, 2019. Significance (p ≤ 0.05) between pairs is denoted by an asterisk next to the group. Shorthand names include East Hudson Strait (EHS), Store Hellefiske Bank (SHB), North Water Polynya (NWP), Davis Strait (DS), Nares Strait (NS), Lancaster Sound (LS), East Barrow Strait (EBS), and Talbot Trough (TT).
(PDF)

**S7 Table. 2021 Surface lipid and fatty acid cluster averages.** Summary averages (±SE) of the seven phytoplankton OceanMet group lipid class percentages, lipid metrics, fatty acid percentages (>1%), and fatty acid biomarkers gathered from surface waters from August 15 – October 4, 2021. If significance (p ≤ 0.05) was found among groups based on a lipid or fatty acid (Table S1), Tukey comparisons in the form of lowercase letters are displayed. Some fatty acids and lipids were not detected (ND). Shorthand names include East Hudson Strait (EHS), North Water Polynya (NWP), Davis Strait (DS), Lancaster Sound (LS), East Barrow Strait (EBS), Beaufort Sea (BF), and Canadian Arctic Archipelago (CAA).
(PDF)

**S8 Table. 2021 Surface PERMANOVA results.** Permutational analysis of variance (PERMANOVA) pairwise results among the nine OceanMet groups (shorthand area name) created from phytoplankton gathered from surface waters from August 15 – October 4, 2021. Significance (p ≤ 0.05) between pairs is denoted by an asterisk next to the group.
(PDF)

**S9 Table. 2021 Correlations.** Regression summary correlating the oceanic measurements (bottom depth, temperature, salinity, light transmission, dissolved oxygen, oxygen saturation, nitrate sensor, nitrate + nitrite, silicate, phosphate, and ammonium) with lipid class percentages, lipid biomarkers, fatty acid percentages, and fatty acid biomarkers gathered from surface (A) and sub-surface chlorophyll maximum (SCM)(B) waters between August 15 – October 4, 2021 from the East Hudson Strait, through Baffin Bay and the Canadian Arctic Archipelago, ending in the Beaufort Sea. Lipid and fatty acid classes presented are above, on average amongst all samples, 1% of the phytoplankton gathered from all locations in both 2019 and 2021. If a correlation was significant (p ≤ 0.05), the correlation coefficient was highlighted as follows: 1–0.5 = orange, 0.49–0 = yellow, −0.01 – −0.49 = light green, −0.5 – −1 = green. *Ammonium was only taken from the Beaufort Sea and Canadian Arctic Archipelago stations, so results only reflect correlation with those stations. ^Light transmission and dissolved oxygen were only taken from the East Hudson Strait, Baffin Bay, and the eastern Canadian Arctic Archipelago stations, so results only reflect correlations with those stations.
(PDF)

**S10 Table. 2021 SCM lipid and fatty acid cluster averages.** Summary averages (±SE) of the five phytoplankton OceanMet group lipid class percentages, lipid metrics, fatty acid percentages (>1%), and fatty acid biomarkers gathered from sub-surface chlorophyll maximum (SCM) waters from August 15 – October 4, 2021. If significance ($p \leq 0.05$) was found among groups based on a lipid or fatty acid (Table S1), Tukey comparisons in the form of lowercase letters are displayed. Some fatty acids and lipids were not detected (ND). Shorthand names include East Hudson Strait (EHS), North Water Polynya (NWP), Davis Strait (DS), Lancaster Sound (LS), East Barrow Strait (EBS), Beaufort Sea (BF), and Canadian Arctic Archipelago (CAA). (PDF)

**S11 Table. 2021 SCM PERMANOVA results.** Permutational analysis of variance (PERMANOVA) pairwise results among the nine OceanMet groups (shorthand area name) created from phytoplankton gathered from sub-surface chlorophyll maximum (SCM) waters from August 15 – October 4, 2021. Significance ($p \leq 0.05$) between pairs is denoted by an asterisk next to the group.
(PDF)

## Acknowledgments

We are all thankful to the officers and crew of the CCGS (NGCC) *Amundsen*, to Jonathan Gagnon for field sampling in 2019, and to Gabrièle Deslongchamps, Jorge A. del Ángel-Rodríguez, and Jeanette Wells for laboratory analysis and technical support. We thank Pascal Guillot for CTD data processing, and also the open-source software (Ocean Data View). CS conducted field sampling in 2021, performed the laboratory and statistical analysis, and wrote the manuscript. CCP and J-ÉT created the project, acquired funding, designed the sampling, and revised the manuscript. All authors contributed to the article and approved the submitted version. We would also like to thank Jens Nielsen, Ian Fleming, and Paul Snelgrove for reviewing a version of this manuscript that was submitted as a thesis chapter.

## Author contributions

**Conceptualization:** Carlissa Salant.

**Data curation:** Carlissa Salant.

**Formal analysis:** Carlissa Salant.

**Funding acquisition:** Christopher C. Parrish, Jean-Éric Tremblay.

**Investigation:** Carlissa Salant, Christopher C. Parrish, Jean-Éric Tremblay.

**Methodology:** Carlissa Salant.

**Project administration:** Jean-Éric Tremblay.

**Resources:** Christopher C. Parrish, Jean-Éric Tremblay.

**Supervision:** Christopher C. Parrish.

**Validation:** Carlissa Salant.

**Visualization:** Carlissa Salant.

**Writing – original draft:** Carlissa Salant.

**Writing – review & editing:** Carlissa Salant, Christopher C. Parrish, Jean-Éric Tremblay.

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
