## [Decision Letter · Decision Letter 0]

16 Sep 2025

Dear Dr. Salant,

Thank you for submitting your manuscript to PLOS ONE. I have received two detailed reviews and I think you will find these constructive reviews helpful in revising the paper. At this point, the manuscript is not suitable for acceptance, but I invite you to submit a revised version of the manuscript that addresses the points raised during the review process. Both reviewers provide detailed critiques; Reviewer 2 for example was concerned about putting prior work such as Marmillot et al. (2020) in better context---it is only briefly mentioned, and also providing  a more well developed conclusion. Reviewer 1 provided a number of line by line comments as well as expressing concern about re-organizing the Results section so it is more streamlined. Both reviewers have read the paper in more detail than I have, but I very quickly saw minor "polishing" issues such as repeatedly defining the North Water Polynya (as NWP), when that should only be necessary once, and then also spelling out North Water Polynya rather than using the NWP definition you have provided. And using variant spellings of polynya/polynia (I think polynya is clearly preferred). Despite these issues, both reviewers had a positive view of the value of your data set, so please consider their constructive criticisms in a positive light as you revise the paper.

We look forward to receiving your revised manuscript.

Kind regards,

Lee W Cooper, Ph.D.

Section Editor

PLOS ONE

Journal Requirements:

This investigation was supported by Memorial University of Newfoundland, a grant from the Natural Sciences and Engineering Research Council of Canada (NSERC) to CCP and J-ÉT via the strategic network CHONe (Canadian Healthy Oceans Network), and a grant from the network center of excellence ArcticNet to J-ÉT and CCP.

We are all thankful to the officers and crew of the NGCC Amundsen, to Jonathan Gagnon for field sampling in 2019, and to Gabrièle Deslongchampes, Jorge del Angel, and Jeanette Wells for laboratory analysis and technical support. We thank Pascal Guillot for CTD data processing, and also the open-source software (Ocean Data View). CS conducted field sampling in 2021, performed the laboratory and statistical analysis, and wrote the manuscript. CCP and J-ÉT created the project, acquired funding, designed the sampling, and revised the manuscript. All authors contributed to the article and approved the submitted version.

This investigation was supported by Memorial University of Newfoundland, a grant from the Natural Sciences and Engineering Research Council of Canada (NSERC) to CCP and J-ÉT via the strategic network CHONe (Canadian Healthy Oceans Network), and a grant from the network center of excellence ArcticNet to J-ÉT and CCP.

This investigation was supported by Memorial University of Newfoundland, a grant from the Natural Sciences and Engineering Research Council of Canada (NSERC) to CCP and J-ÉT via the strategic network CHONe (Canadian Healthy Oceans Network), and a grant from the network center of excellence ArcticNet to J-ÉT and CCP.

5. Thank you for uploading your study's underlying data set. Unfortunately, the repository you have noted in your Data Availability statement does not qualify as an acceptable data repository according to PLOS's standards.

6. In the online submission form, you indicated that all relevant lipid and fatty acid data are within the manuscript and its supporting Information files. All oceanographic files are available from the Amundsen Science database (https://amundsenscience.com/data/data-access/). All nutrient files are available upon request from Tremblay, JET.

7. We note that Figure 1 in your submission contain map images which may be copyrighted. All PLOS content is published under the Creative Commons Attribution License (CC BY 4.0), which means that the manuscript, images, and Supporting Information files will be freely available online, and any third party is permitted to access, download, copy, distribute, and use these materials in any way, even commercially, with proper attribution. For these reasons, we cannot publish previously copyrighted maps or satellite images created using proprietary data, such as Google software (Google Maps, Street View, and Earth). For more information, see our copyright guidelines: http://journals.plos.org/plosone/s/licenses-and-copyright.

9. Please remove all personal information, ensure that the data shared are in accordance with participant consent, and re-upload a fully anonymized data set.

Additional guidance on preparing raw data for publication can be found in our Data Policy (https://journals.plos.org/plosone/s/data-availability#loc-human-research-participant-data-and-other-sensitive-data) and in the following article: http://www.bmj.com/content/340/bmj.c181.long .

Reviewers' comments:

Reviewer's Responses to Questions

**Comments to the Author**

1. Is the manuscript technically sound, and do the data support the conclusions?

Reviewer #1: Yes

Reviewer #2: Partly

2. Has the statistical analysis been performed appropriately and rigorously?

Reviewer #1: Yes

Reviewer #2: Yes

3. Have the authors made all data underlying the findings in their manuscript fully available?

Reviewer #1: Yes

Reviewer #2: No

4. Is the manuscript presented in an intelligible fashion and written in standard English?

Reviewer #1: No

Reviewer #2: No

Reviewer #1: This study examines lipid classes, fatty acid markers, and phytoplankton assemblage metrics to evaluate oceanographic drivers of lipid content, composition, and quality across years in the Canadian Arctic. The dataset is valuable, and the study addresses important ecological questions regarding lipid synthesis and phytoplankton quality. However, the manuscript in its current form reads more like a draft than a publication-ready paper. The text is overly long, with the Results in particular providing more narrative than necessary. I recommend streamlining the manuscript by focusing on the key findings and using the figures to illustrate the results rather than explaining them in detail. I have provided specific comments and suggestions throughout that, if addressed, should strengthen the clarity and impact of the study.

Line 38: Define DHA/EPA or note in parentheses that these are the marine omega 3 FAs

Line 72-86: Nice summary

Line 107: This should be North Water Polynya (not North Way Polynia). Also polynya is spelled incorrectly.

Line 119: condition may be fine alone, without “healthfulness”. Or perhaps nutritional quality would be better

Line 122-123: Do you have a reference for this statement?

Line 124 coastal or riverine/fluvial/terrestrial

Line 139: Italicize Amundsen

Line 141: Seawater for both filtered phytoplankton and nutrient analyses

Line 144-145: Change to “Filters were pre-combusted at 450 °C for 12 h and stored in aluminum foil.”

Line 146: Also -20 freezer at the lab?

Line 146-153: I’m confused by the statement and maybe just not understanding as written. Why were three filters combined for each station in 2021 and only one for 2019?

Line 185: missing bracket/reference

Line 190-191: Awkward sentence. Revise to “The extracts were applied to silica gel–coated Chromarods using a micropipette.”

Line 191-192: Perhaps scrap previous sentence and start with “Lipids were separated using a four-step development procedure on silica gel Chromarods.”

Line 191-198: Do you have a reference for this method that you followed or adapted?

Line 201: pre-combusted

Line 201-202: volume of H2SO4-MeOH added?

Lines 213-217: Skewness not skewedness. Also note the tests used (examples in parentheses). Consider rephrasing to: “Twelve hydrographic and chemical variables (temperature, salinity, bottom depth, fluorescence, light transmission, oxygen saturation, dissolved oxygen, nitrogen probe, nitrate, silicate, phosphate, and ammonium) were evaluated for skewness (e.g. Shapiro–Wilk test), homogeneity of variances (e.g. Levene’s test), and collinearity (variance inflation factors, VIF) prior to analysis. Variables with skewed distributions were log-transformed. Concentration variables were also log-transformed to stabilize variances before statistical analyses.”

Line 227: healthfulness health? As above, consider ‘quality’ or ‘nutritional quality’ instead?

Line 229-231: Phrasing is a bit unclear. I would break them up rather than list 1-11. Ex: Metrics (of what?) used included total lipids, TAG/PL, TAG/ST. FA biomarkers included PUFA/SFA, Σω3, Σω6, and DHA/EPA. Phytoplankton assemblage markers were classified as bacterial, diatom, flagellate, or coastal margin.

Line 231-232: Explain why you do this. DHA and EPA are the marine-sourced omega-3 FAs. Did this differ much from the Σω3?

Line 240: replace “is also discussed” to “were also assessed for…”

Line 244 and 248, 249, 251: revise “oceanic” to “oceanographic”

Line 256: “Arctic area” sounds strange since the entire study is in the Canadian Arctic. Suggest changing to “sites” or “stations”…. “although sometimes our resulting cluster groups were from the same stations.”

Line 258: Same as above. Reconsider use of “Arctic areas”

Lines 212-279: The statistical analyses section is too dense and written as narrative. This section could be streamlined. I have tried to offer some suggested revised text below that incorporates some of my earlier comments:

“Twelve hydrographic and chemical variables (temperature, salinity, bottom depth, fluorescence, light transmission, oxygen saturation, dissolved oxygen, nitrogen probe, nitrate, silicate, phosphate, and ammonium) were evaluated for skewness (e.g. Shapiro–Wilk test), homogeneity of variances (e.g. Levene’s test), and collinearity (variance inflation factors, VIF) prior to analysis. Variables with skewed distributions were log-transformed. Concentration variables were also log-transformed to stabilize variances before statistical analyses.

Total lipids are reported in mg g⁻¹ wet weight (WW), while lipid classes and fatty acids (FAs) are expressed as proportions (% total). Analyses were restricted to lipid classes and FAs contributing >1% of the total across both years, yielding eight lipid classes (of 14 possible) and 17 FAs (of 74 identified). In addition to individual groups, we used three lipid ratios (total lipids, TAG/PL, TAG/ST) and eight FA-based biomarkers (PUFA/SFA, Σω3, Σω6, DHA/EPA, bacterial, diatom, flagellate, coastal margin); DHA+EPA was also included for comparison despite overlap with Σω3.

Analyses were conducted separately for four groups defined by year and depth (2019 surface, 2019 SCM, 2021 surface, 2021 SCM). Correlation analyses were performed between all hydrographic variables and lipid classes, FAs, and biomarkers, and the number of significant correlations across groups was summarized.

Multivariate analyses were performed in PRIMER v7. Hierarchical cluster analyses with similarity profile tests (SIMPROF, Type 1, 999 permutations) were conducted for each year and depth using Euclidean distance. Outliers identified by SIMPROF were excluded. Final clusters, based on oceanographic properties only, were termed “OceanMet” groups; nutrient profiles were examined subsequently in relation to these groups. One-way PERMANOVA (Type III SS, 999 permutations) with Bray–Curtis similarity was used to test for differences among OceanMet groups, followed by pairwise tests. Principal coordinates analysis (PCO) was used to visualize relationships between OceanMet groups, lipid metrics, and biomarkers (TAG/ST excluded due to missing values), using log(x+1)-transformed data and Bray–Curtis similarity.

Finally, univariate comparisons of lipid classes and FAs among OceanMet groups were tested using one-way ANOVA with Tukey’s HSD or, when assumptions of normality/variance were violated, non-parametric Kruskal–Wallis tests (Minitab v21).”

Line 281–288: This section reads more like the methods/site description. Remove from results.

Line 311: It is unnecessary to keep qualifying observations as “ocean” or “oceanic”. This is known. Just say the inorganic nutrient measurements.

Line 326-329: FAs have already been defined. Perhaps SFA, MUFA and PUFA should also be defined in the introduction and their relevance to quality indicators.

Results:

The Results section could be streamlined by using the figures to illustrate findings, rather than providing lengthy verbal descriptions of each panel. This would make the narrative more concise and easier to follow. Consider tightening the Results by highlighting the key patterns and directing the reader to the relevant figures, rather than describing each figure in detail.

Line 656: question mark in there that shouldn’t be

Figures:

Line 391: Replace “mainly negative” with X out of X being negatively correlated. Suggest being more specific in the others too (Lines 402-405)

Lines 448-449: This is more discussion

Line 470-472: Sentence is clunky and wordy. Suggest “Fluorescence was generally negatively correlated with coastal FA and biomarker signatures, and positively correlated with diatom-associated FAs and biomarkers.”

Lines 488-496: As before, this section should be incorporated in the methods/site description, or scrapped entirely. We don’t need a descriptive account of the cruise tracks.

Discussion:

Lines 732-742: Consider eliminating this paragraph. You don’t need to tell us what you are about to tell us. Just tell us.

Line 765: refer the reader back to figure (Fig 2B)

Line 810: same as above, point to figure

Lines 818-819 and Line 950: Don’t DHA/EPA positively correlate with temperature in Fig 4? Or is DHA/EPA different from DHA + EPA?

Line 829: Reference style differs from the rest of the paper (numeric)

Lines 850-851: Is that what you see in your dataset?

Lines 876-887: Why are we talking about spring blooms in September-October? It seems reasonable to me that dinoflagellates would have dominated the phytoplankton community at this time of year over diatoms. Was sample collection date assessed to have any statistical significance? It would also be interesting to have a metric related to sea ice break up date or some other indicator of length of time from spring bloom to sampling date.

Line 913: Consider reviewing Ardynya and Arrigo (2022) review: https://www.nature.com/articles/s41558-020-0905-y There is some information in here about relationships between nutrients and phytoplankton communities

Line 924-928: My guess is that sample collection date is the important factor here.

Figures:

Figure 1: “polynia” should be polynya

Figure 2: Reconsider the phrasing to describe the Tukey comparisons and letters used in the figure. I think I know what you mean here but it could be better explained in the figure caption.

Consider combining Figs 2 and 5 and Figs 3 and 6 for ease of comparison between years.

Figure 4: I like this figure a lot. Much clearer to understand the bigger picture. Caption refers to Supplemental Tables X-X?

Reviewer #2: Review:

Phytoplankton fatty acid proportions in the Canadian Arctic are strongly affected by temperature, salinity, and phosphate in late summer

Carlissa D. Salant, Jean-Éric Tremblay, Christopher C. Parrish

The authors have produced an impressive dataset on plankton lipid classes and fatty acid compositions from the Canadian Arctic, covering a wide range of stations, two different sampling years and two depth strata. The data were linked to a range of environmental predictors (e.g. temperature, light, nutrient concentrations) to understand the potential changes in nutritional quality at the base of the Arctic food web. Even though the data are highly valuable, and the aim of the study is well defined, the manuscript requires substantial improvement.

There are two major comments:

1. The manuscript shows strong overlap with the study by Marmillot et al. (2018), in terms of sampling region and the range of lipid and environmental data that have been sampled, and also the type of analysis that has been applied to illustrate the results. Therefore, it is surprising that the authors did ‘almost’ ignore the Marmillot study in the Introduction and Discussion, only citing it with other studies in a different context, instead of expanding on the similarity of this particular work. It is important that the authors clearly indicate how this new study compares and extends from Marmillot et al. in both the Introduction and the Discussion. I suggest that key data are directly compared in an additional plot, to either illustrate further interannual differences and/or regional consistency across multiple years of sampling. (see also Nielsen et al 2023, https://doi.org/10.1016/j.dsr2.2022.105247)

2. I rarely came across a manuscript that sticks so close to the actual data and does open up so little to wider considerations. This is to the extent that statistical results are already presented in the Abstract and the manuscript-specific term ‘OceanMet groups’ still occurs 6 times in the Conclusion. On Line 262, the authors promised ‘Overall conclusions, however, are discussed in more broad terms’, but I can see no evidence. Can we foresee if a warmer, ice-free Arctic will have more nutritious algae? And, if not, which additional information would we required? A Conclusion is usually aimed at a wider audience that wants to find the key message without going through the nitty-gritty details of the study. Also, the end of the Discussion or Conclusion is a good place to get back to the study hypotheses and look at the bigger picture – e.g. the future health and biodiversity of the Arctic food web or potential impacts on food provision for local communities/fisheries. Do your study results give any potential insights towards such topics that could create a take-home message for people that are not explicitly interested in biomarker fatty acids?

Other comments:

3. I’m not sure why oxygen concentrations or saturations have been included as a parameter that potentially drives the lipid- and fatty acid composition of the phytoplankton. In the manuscript, I read (Line 904): ‘Dissolved oxygen in the marine environment can limit primary production since it is coupled to the biogeochemical cycle of both nitrogen and phosphorus’. As these nutrients are measured directly, I would not include oxygen concentrations or saturations here, except the authors can provide evidence that oxygen saturations are potentially too low or too high in the Canadian Arctic for the production of certain lipids (so a direct driver of their composition).

4. The authors did not name the biomarker fatty acids for the groups they want to trace (Line 231): 8) bacterial, 9) diatom, 10) flagellate, and 11) coastal margin. Different studies use different fatty acids as biomarkers, e.g. 16:1(n-7) or 16:4(n-1) or 20:5(n-3) for diatoms. This has to be clarified on Line 231, and in the PCA figures. Moreover, it has to be stated which fatty acids have been used to trace vascular plants and green macroalgae (Line 853), to distinct between dinoflagellates and haptophytes (Line 870), and between dinoflagellates and flagellates (Line 883).

5. The statement (Line 79): ’In general, with an increase in temperature, there is an increase in TAG, SFA and MUFA proportions in phytoplankton [24–26].’ needs clarification. If it refers to cultured algae with temperature as a sole driver, there is supporting evidence, but if it refers to field studies in the Arctic then the statement is wrong. In the Arctic, temperature effects are masked by the lack of nutrients and light (see Schmidt et al. 2024, DOI: 10.1111/gcb.17090). Thus, EPA and DHA proportions were lower in the colder Central Arctic Ocean than in the warmer Arctic shelf regions, and in the

Bering Sea and North Bering- and Chukchi Sea the DHA proportions significantly increased with temperature (while EPA proportions did not change) (see Schmidt et al. 2024, Supplement Fig. S5). This matches observations in the present study (see Line 830 and 892) and needs to be cited in this context as it supports accumulating evidence that the temperature-to-PUFA relationship is in the Arctic regions more complex than suggested by Hixon and Arts (2016) based on global data sets.

6. Some labels on the PCA plots (Fig. 3 and Fig. 6) are overlaying and therefore not readable.

7. Some wording is imprecise, e.g. Line 88: ‘Nitrogen can limit primary production….’It’s the lack of nitrogen, I assume. Or Line 138: ‘Phytoplankton samples were collected …’ What you sample with a GF/C filter is particulate matter that can also include sediment, hetero- and mixotrophs. These are only examples, and I suggest you check the text carefully again for your wording.

8. There are several sentences with an unclear message, which might come from using AI. The following just gives a few examples, but I would suggest the whole text is checked by a native English-speaking human for clarity.

Line 721: energetics

Line 747: in regard for correlating with hydrographic properties

Line 751: be different based proximity

Line 755: Similarly, the 2019 SCM OceanMet group differences were due to a high proportion of the polar lipid class in the high temperature/low oxygen saturation, nitrogen, phosphate OceanMet group (all Store Hellefiske Bank stations) and a low proportion in the high oxygen saturation/low salinity OceanMet group (East Barrow Strait and North Water Polynya stations).

Line 767: total lipids was

Line 878: temperature phytoplankton

Line 878: the increase in the expanding

Line 895: Given the narrow range within the Arctic for both the surface and SCM waters

Line 902: health phytoplankton

Line 781: As a limiting nutrient in the Arctic related to the magnitude of phytoplankton blooms [45,74,75]

**Do you want your identity to be public for this peer review?** For information about this choice, including consent withdrawal, please see our Privacy Policy

Reviewer #1: No

Reviewer #2: No

---

## [Editor Report · Decision Letter 1]

8 Dec 2025

Dear Dr. Salant,

Thank you for submitting your revised manuscript to PLOS ONE, and congratulations on completing your Ph.D. degree. I think you have made a good effort responding to the two reviewers and I don't think it is necessary that I return the manuscript to them for any additional evaluation. However, I have had the opportunity to look over the manuscript myself and I think there are some remaining improvements that can be made. For the most part, my suggestions are editorial and grammatical, rather than addressing major scientific questions. But in the interests of improving the communication of your research results, I am inviting you to submit a revised version of the manuscript that addresses the points and suggestions that I make below.

Line 43. “was” is crossed out, but it seems necessary to the grammatic al structure of the sentence---otherwise there is no verb in the sentence.

Line 55-56 This sentence is grammatically faulty as now written: “‘They act as the base for the transfer of energy, carbon, and essential nutrients are transferred to primary consumers and subsequently to predators” Transfer appears twice, and two verbs, “act” and “are”

Line 58-62. I would add “studies“ at two points in this sentence to make the sentence clearer: While some <studies> suggested no initial deleterious effect on primary consumer health with decreased essential nutrient value of primary producers [13,14], other <studies> have suggested that over the long term, a decrease in primary producer essential nutrient value would negatively affect consumer health at multiple food web levels [15,16].”

Line78-84. I found these two sentences to be confusing. You mention three seasons, spring, summer and winter in the two sentences and the controlling factors for productivity, but in the end it is not clear to me which season is the point of the sentence.

Line 101. I would add “cell” after individual

Line 106. Concentrations would be a better word here than values.

Line 120-121. I would change various to varying, and on the next line drop the word amounts.

Line 129-130. On line 129, you use the verb “compiled” in the past tense and on the next line in the line you use “investigate” in the present tense. Do settle on one tense and keep it consistent. The text that follows seems to be mostly in the past tense.

Line 140, Line 1034. A comma generally follows i.e. in American English, although not in British English, so I would say that this is your call.

Line 154. Use among here instead of between if there are more than two comparisons being made.

Line 162. Retried should be retrieved.

Line 163. To say that biogeochemical sensors collected water isn’t quite right.

Line 164 The word water appears twice in a row in the sentence.

Line 192. The commercial vendor here is referred on this line as both Sea Bird and Sea-Bird. The company’s name is Sea-Bird Scientific. I’d suggest using the full name of the company and to keep it consistent throughout the text.

Line 194. What is meant by a nitrogen probe? Is this a nitrate sensor such as the Sea Bird instrument that is marketed commercially? Could you please change this so it is clear?

Line 196. Add “filter holder” after Swinex. I think alternatively, you could just say that the water samples were filtered through GF/F filters. The mechanics of how that was done and the supplies used are not that critical.

Line 197-198. Again, I am confused by the terminology of nitrogen probe, and since it appears to correspond to nitrate, call it a nitrate sensor. Likewise, a fluorescence measurement is made in voltage units. It can be converted if calibrated to chlorophyll fluorescence in mg/m^3, but the way this is expressed is not very clear.

Line 204. Add “that” after “noting”

Line 212-213. Usually the format for a personal communication, is to state the name of the person communicating this information, then the date, and then the words, uncapitalized, personal communication

Line 214. A comma is not needed after pre-combusted

Line 219. The article “a” appears twice consecutively in this sentence.

Line 232. Since the subject of the sentence is plural (stages), contains should be contain and separates should be separate.

Line 238. mL is preferred over ml, but both are used in the manuscript. Please do a universal search and make all uses of milliliter consistent.

Line 247. Here you use the manufacturer name, and city, state abbreviation and country location, but elsewhere you only give the name of the manufacturer and not where the company is located, e.g. Sea-Bird. Please make this consistent throughout.

Line 251. Nitrogen probe is not a variable

Line 261. “in” should be “as”

Line 301. I would replace “not nutrients” with “excluding nutrients”

Line 327. PCO should be PCA

Line 456. I am used to seeing principal components analysis abbreviated as PCA, but I suppose it is OK if you consistently use PCoA

Line 488, 500, 504, 676, 801.  Still have this question as to what the nitrogen probe means

Line 569. Given you are discussing multiple group comparisons, “among” is preferable to “between”

Line 583. “lipids was” should be “lipids were”

Line 585. Don’t think the commas on either side of generally are needed.

Line 596. Again, I don’t think commas are needed on both sides of “positively”

Line 679, 680. Instead of “taken” use “made” on line 679, and “determined” on line 680.

Line 701, 720, 730, 803, 830, 836, 842. Use among rather than between for comparison involving multiple (>2) groups.

Line 725, 840. The mention of both groups and OceanMet groupings, separated by a comma is confusing. Are the two terms connected or separate?

Line 730, 844. Since you have already provided a definition for the abbreviation PCoA, you don’t have to provide it again here.

Line 836. Change “no significance” to “no significant difference”

Line 888. Complimentary should be complementary

Line 894. “in” doesn’t sound quite right to me---I would change to on, which is more often used with focus.

Line 913. Change “which” to “that”

Line 967. Replace “had” with “having”

Line 1013. Delete “that”---it introduces a grammatical error into the sentence.

Line 1023. Not sure I understand what summations means here.

Line 1062. Suggests should be suggest since the subject of the sentence, markers, is plural.

Line 1075. Temperature should be temperate

Line 1089. For this study sounds better than from this study.

Line 1093. Your data “suggest” a correlation, but can’t you be more definitive than that? A correlation is either significant or it isn’t.

Line 1096-1098. This sentence isn’t clear. Which study is being referred to in the sentence (“this study”)---the study you referenced in the prior sentence or your own study?

L:ine 1105. The wording here is odd. Maybe unhealthy organisms should be stressed organisms?

Line 1109. Do you mean the lack of dissolved oxygen?

Line 1111. Low oxygen levels?

Line 1115. You use nitrogen repeatedly throughout the manuscript, but you really mean nitrate + nitrite (and perhaps ammonia). Also, here you use the element name phosphorus, but you really mean phosphate. I would try to use the actual chemical form as much as possible rather than the elemental name. Possibly using nitrogen is OK, since it takes several forms, but phosphate is the form of that element that is encountered in seawater.

Line 1193. Since the polynya is really a geographical area, so you need to add “area” after it?

Line 1194. Should “nothing” be “noting”? Probably also need a comma after productivity.

Line 1196, 1200. Brackets rather than parenthesis for the references.

Line 1203. Add “that” before “have”

Line 1206. Do you distinguish the North Water Polynya as not being in the Canadian Arctic Archipelago.

Line 1235 I am not sure what a high lipid phytoplankton condition is. Please make this sentence clearer.

Line 1238. You are using the French language initials for the registry of the Amundsen, which I think maybe OK, but PLOS is an English language journal so using the initials CCGS instead of NGCC may be more consistent with the journal guidelines.

Line 1241. There is a specific reference recommendation for citing Ocean Data View, from the software license agreement, e.g., Schlitzer, Reiner, Ocean Data View, https://odv.awi.de, 2022. This is best done in the figure caption where you use the software, then add the reference to Schlitzer, 2022 to your reference list

Line 1358-1362, 1416-1417, 1543-1544, 1594-1595. All words in the titles are capitalized so it isn’t consistent with your other references.

Line 1390. Italicize Synechococcus

Line 1498, 1517. Italicize the species name in the title.

Line 1535. Capitalize Atlantic

That's all I have, and despite being a long list, I judge these recommended changes to constitute minor revisions.

Could you please submit your revised manuscript by Jan 22 2026 11:59PM. If you will need more time than this to complete your revisions, please reply to this message or contact the journal office at plosone@plos.org . A rebuttal letter that responds to each point that I raised in my role as the academic editor. You should upload this letter as a separate file labeled 'Response to Editor'.A marked-up copy of your manuscript that highlights changes made to the original version. You should upload this as a separate file labeled 'Revised Manuscript with Track Changes'.An unmarked version of your revised paper without tracked changes. You should upload this as a separate file labeled 'Manuscript'.

I look forward to receiving your revised manuscript.

Kind regards,

Lee W Cooper, Ph.D.

Section Editor

PLOS One
---

## [Author Response · Author response to Decision Letter 2]

16 Dec 2025

Please see attached document for line-by-line responses to the most recent Editor comments from December 8th, 2025.

---

## [Editor Report · Decision Letter 2]

21 Dec 2025

Phytoplankton fatty acid proportions in the Canadian Arctic are strongly affected by temperature, salinity, and phosphate in late summer

PONE-D-25-43058R2

Dear Dr. Salant,

Thank you for responding to my letter and revising the paper to reflect the final editing suggestions. I'm pleased to inform you that your manuscript has been judged scientifically suitable for publication and will be formally accepted for publication once it meets all outstanding technical requirements.

Kind regards,

Lee W Cooper, Ph.D.

Section Editor

PLOS One

---

## [Editor Report · Acceptance letter]

PONE-D-25-43058R2

PLOS One

Dear Dr. Salant,

I'm pleased to inform you that your manuscript has been deemed suitable for publication in PLOS One. Congratulations! Your manuscript is now being handed over to our production team.

Kind regards,

on behalf of

Dr. Lee W Cooper

Section Editor

PLOS One